# Linking diet switching to reproductive performance across populations of two critically endangered mammalian herbivores

Nick Harvey Sky [1,2] ✉, Jake Britnell [1,2], Rachael Antwis [3], Tyler Kartzinel [4,5], Daniel Rubenstein [6], Phil Toye [7], Benedict Karani [7], Regina Njeru[7], Danielle Hinchcliffe[8], Jamie Gaymer [9], Samuel Mutisya[10] & Susanne Shultz [1]

Optimal foraging theory predicts that animals maximise energy intake by consuming the most valuable foods available. When resources are limited, they may include lower-quality fallback foods in their diets. As seasonal herbivore diet switching is understudied, we evaluate its extent and effects across three Kenyan reserves each for Critically Endangered eastern black rhino (*Diceros bicornis michaeli*) and Grevy's zebra (*Equus grevyi*), and its associations with habitat quality, microbiome variation, and reproductive performance. Black rhino diet breadth increases with vegetation productivity (NDVI), whereas zebra diet breadth peaks at intermediate NDVI. Black rhino diets associated with higher vegetation productivity have less acacia (Fabaceae: *Vachellia* and *Senegalia spp*.) and more grass suggesting that acacia are fallback foods, upending conventional assumptions. Larger dietary shifts are associated with longer calving intervals. Grevy's zebra diets in high rainfall areas are consistently grass-dominated, whereas in arid areas they primarily consume legumes during low vegetation productivity periods. Whilst microbiome composition between individuals is affected by the environment, and diet composition in black rhino, seasonal dietary shifts do not drive commensurate microbiome shifts. Documenting diet shifts across ecological gradients can increase the effectiveness of conservation by informing habitat suitability models and improving understanding of responses to resource limitation.

Species' ranges encompass ecological gradients from core, or optimal, to marginal habitats[1,2]. Optimal habitats are associated with high population density, reproductive rates, survivorship and resilience whereas marginal habitats are associated with lower viability and increased rates of local extirpation[3]. Individual fitness and demographic heterogeneity, or variation in survival and reproduction, scale up to produce these spatial and temporal disparities in population performance across a range[4]. Core areas, with high ecological resilience, are not necessarily near the centre of a species' range[5]. Identifying and prioritising resilient populations, and ecological characteristics that describe optimal habitats, are key for predicting and arresting biodiversity loss[6,7], especially where a species' range is dynamically changing in response to environmental change[8,9]. Thus, a primary challenge for modern conservation biology is developing evidence-based relationships between ecological conditions, individual fitness and demographic heterogeneity across space and time[10].

[1]Department of Earth and Environmental Sciences, University of Manchester, Manchester M13 9NT, UK. [2]North of England Zoological Society, Chester Zoo, Upton-by-Chester CH2 1LH, UK. [3]School of Environment and Life Sciences, University of Salford, Salford M5 4WX, UK. [4]Department of Ecology, Evolution, and Organismal Biology, Brown University, 85 Waterman Street, Providence, RI 02912, USA. [5]Institute at Brown for Environment and Society, Brown University, 85 Waterman Street, Providence, RI 02912, USA. [6]Department of Ecology and Evolutionary Biology, Princeton University, Princeton, NJ 08544-2016, USA. [7]International Livestock Research Institute and Centre for Tropical Livestock Genetics and Health, Nairobi P.O. Box 30709-00100, Kenya. [8]School of Biological and Environmental Sciences, Liverpool John Moores University, Liverpool L3 3AF, UK. [9]OI Jogi Ltd., PO Box 259-10400 Nanyuki, Kenya. [10]OI Pejeta Conservancy, PO Box 167 Nanyuki, Kenya. ✉e-mail: nick.c.harvey@gmail.com

https://doi.org/10.1038/s42003-024-05983-3     **Article**

Fitness is directly associated with maintaining sufficient energy reserves to support metabolism, invest in reproduction, and buffer periods of scarcity[11]. During periods of scarcity, animals can move to follow changing distributions of valuable foods, alter their foraging strategies, shift their diets, or any combination of those. Dietary changes can cause energetic stress, particularly when animals switch to consuming less preferred fallback foods during times of scarcity[12] that they may not be physiologically adapted to digest[13–15]. Diet-switching often occurs between high and low vegetation productivity periods (i.e., summer/winter or before/after rains) in grazers such as bison (*Bison bison*)[16] as well as mixed-feeders such as moose (*Alces alces*)[17]. Despite its potential importance, the role of dietary strategy in driving herbivore population dynamics is poorly understood[18].

Optimal foraging theory (OFT) can predict the composition of a species' diet under particular conditions. During periods of abundance, herbivore diets should contain relatively few species as animals maximize their energy intake by concentrating on the best food plants available in terms of energy and nutrients. When and where valuable foods are more limited, animals should expand their diets to include lower-quality fallback foods[17,19]. Whilst OFT can help predict what foraging strategy is optimal under a given set of conditions, it cannot indicate the impact that each diet will have on survival or reproductive success. Intraspecific comparisons of the frequency with which diet-switching is required across ranges[20] and corresponding demographic indicators can provide a mechanistic explanation for how ecological conditions lead to demographic heterogeneity. Fallback foods can be identified by heavy use during periods of scarcity, or by research into nutritional qualities[21]. Fallback foods have most often been studied in primates, and examples show that the prevalence of these foods is associated with individual fitness proxies and population performance. Whilst the density of gibbons correlates with the density of their fallback food of figs[21], prolonged use of the corms of grasses and sedges can lead to longer interbirth intervals and aborted foetuses in baboons[22].

Different herbivore guilds face different challenges and opportunities in terms of the taxonomic and phylogenetic diversity of plant foods they can accommodate[23], their exposure toxicity from plant defences[24] and the temporal scales over which staple plant foods flush and maintain nutritious foliage[25,26]. Thus, OFT gives rise to different predictions for how herbivores with different foraging strategies respond to changing resource availability[23] (Fig. 1). Seasonal dietary switching has been linked to higher population abundances in mixed-feeders[18] because these switches can buffer against seasonal scarcity[27]. In grazers, poor resource availability has been linked to poor long-term population growth rates, density, and fecundity[28]. Hence, even temporary, seasonal use of fallback plant families (i.e., non-grasses) by grazers may contribute to poor performance[29]. Switching between grass and browse is limited among browsers[27]. Although they not be able to take advantage of high-quality and high-biomass grass flushes[18], the ability to eat chemically and structurally defended woody plants can mitigate the effects of drought[30].

Diet switching may impose a cost on metabolic efficiency. Changes in diet have knock-on effects on the gastrointestinal microbial community (the microbiome) that performs key functional roles in the digestive tract[31–33], including in herbivore species[20,34]. The cost of switching hypothesis suggests that digestion is less efficient when the microbiome community is in flux as the microbiome becomes temporarily mismatched with diet[35]. The effect of this depends on the suddenness of the dietary shift as well as the length of the lag between diet change and microbiome shifts[36]. Although the links are not well understood, microbial community composition has been associated with reproductive performance[37]. Thus, microbiome characteristics associated with poor diet, and in turn, with poor fitness and population performance[38] can be biomarkers of ecological marginality[39].

One challenge in identifying valuable and fallback foods lies in disentangling whether differences in diet between individuals are due to differences in preference or access[17]. This can be solved through longitudinal studies that

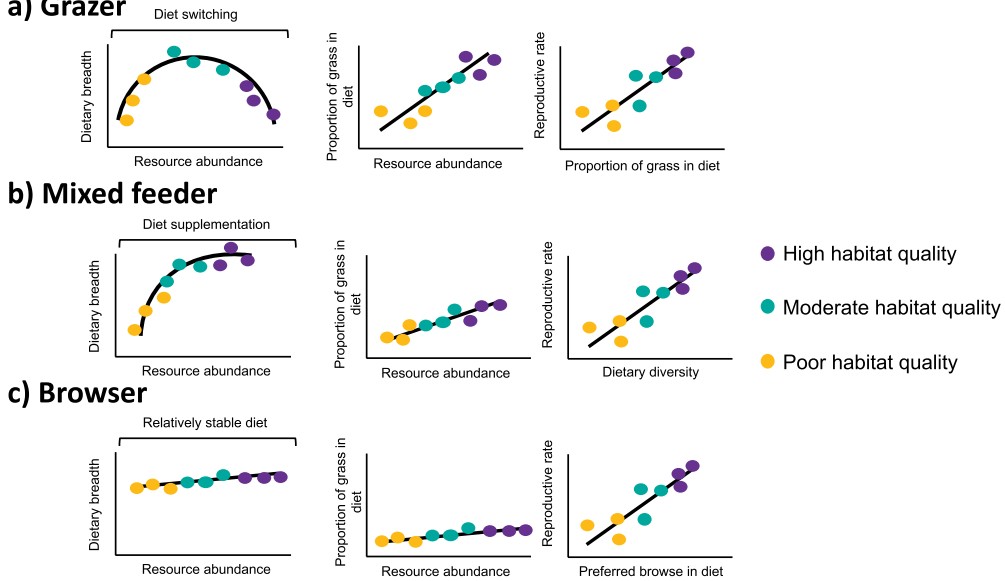

**Fig. 1 | Optimal foraging theory predictions for hypothetical large mammalian herbivores with different feeding strategies.** Optimal foraging theory predictions for a hypothetical: **a** strict grazer, **b** mixed feeder and **c** strict browser. For grazers (**a**), we predict diversity follows a negative quadratic with increasing resource abundance as animals switch to fallback foods during resource-scarce times. As the rains begin and plants start to green up, grazers incorporate some valuable plant taxa and diversity begins to increase. As rains continue grazers transition completely to valuable plant taxa and diversity decreases again. Reproductive rates should positively correlate with the proportion of valuable foods eaten. For mixed feeders (**b**), we predict dietary breadth increases with increasing resource availability, as individuals supplement woody fallback foods, which are available all year-round, with valuable herbaceous foods that are only available after greening up. We predict mixed feeders

do not completely transition to an alternative dietary plant like a grazer. In high-quality areas, high-value food is available alongside fallback foods to allow individuals to survive periods of scarcity. For browsers (**c**), we predict woody plants are most valuable and are available all year-round, although availability changes. Dietary diversity may increase at high resource abundance because they feed opportunistically and there is no need to be selective, and when resources are scarce because they have to take all food that is available and cannot eat too much of certain fallback foods' chemicals. However, overall diet diversity is not predicted to change significantly. Breeding rates are highest in places with a high abundance of valuable woody plants. Yellow dots represent individuals studied in poor habitat, teal dots in moderate habitat quality, and purple dots in high habitat quality.

     

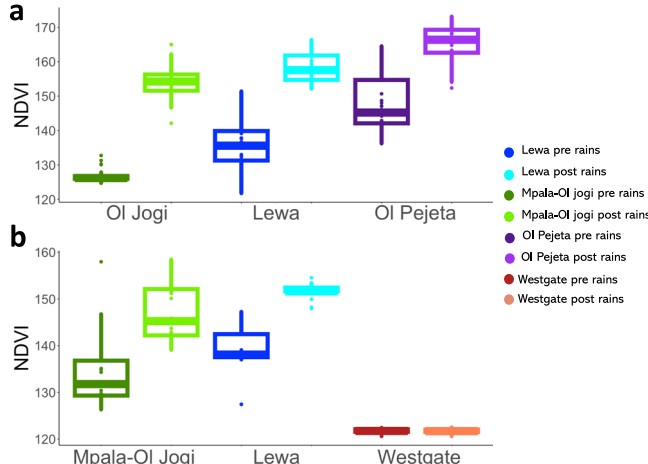

**Fig. 2 | Vegetation productivity varies across reserves and seasons.** Variation in NDVI across reserves and sampling periods for **a** black rhino ($n = 214$) and **b** Grevy's zebra ($n = 154$). The lower, middle and upper horizontal lines show the 25th percentile (lower hinge), and mean and 75th percentile (upper hinge) respectively. Whiskers extend to the largest and smallest values no further than 1.5 times the inter-quartile range away from the hinges. Data points beyond the whiskers are shown as points. Dark blue represents samples taken on Lewa pre-rains, and light blue post-rains. Dark green represents samples taken on Mpala or Ol Jogi pre-rains, and light green post-rains. Dark purple represents samples taken on Ol Pejeta pre-rains, and light purple post-rains. Dark red represents samples taken on Westgate pre-rains and orange post-rains.

focus on how diet changes between lean and plentiful periods across ecological gradients. East African savanna ecosystems present an excellent opportunity to evaluate seasonal diet switching because rainfall is seasonally concentrated such that the diversity and biomass of both herbaceous and deciduous plants increase with the onset of rains[40,41]. Green herbaceous above-ground biomass can fall to very low levels in the dry season[42] whilst woody plants and non-deciduous leaves are still present. In the study region, *Euclea divinorum* (Ebenaceae family) is evergreen and *Vachellia drepanolobium* trees (Fabaceae family) keep some of their leaves during dry seasons[43–45]. We will refer to *Vacehllia* and *Senegalia* species as acacia.

We evaluated temporal diet switching and microbiome variation across five reserves containing two IUCN Red List Critically Endangered savanna herbivores with different foraging strategies; the eastern black rhino (*Diceros bicornis michaeli*) and Grevy's zebra (*Equus grevyi*). Savanna herbivore diets can be described along three axes using the proportions of legumes (Fabaceae family), grasses (Poaceae family) and all other families (e.g., trees and shrubs from *Ebenaceae*)[46]. Whilst black rhino diets do vary seasonally[47,48] they are considered to be browsers, to find woody plants in the family Fabaceae particularly valuable[49,50], and to ideally have herbaceous plants, including grasses, contribute little to their diets[47]. Grevy's zebra, in contrast, are grazers with a preference for grasses and other herbaceous vegetation[51,52]. We do not know, firstly the degree to which animals incorporate fallback foods seasonally and secondly, whether dietary switching has implications for fitness. We seek to provide evidence relevant to both questions.

We used DNA metabarcoding to characterise variations in diet and microbiome between several reserves in Laikipia and Meru Counties in Kenya to evaluate our predictions of diet switching before and after rains in different reserves. Stable isotope studies, which are often used to study dietary switching, do not demonstrate dietary changes within C3 and C4 plant groups. The aims of this study were two-fold. First, to evaluate the levels and extent of dietary shifts by black rhino and Grevy's zebra in response to seasonal changes in resource availability across a regional climatic gradient. Second, to determine whether dietary shifts and the use of fallback foods are associated with variation in habitat quality, variation in the microbiome or calving rates in these large herbivores.

## Results

In order to connect seasonal dietary shifts across an environmental gradient to changes in the microbiome and fitness we carried out four main analyses. For each sampled individual we characterised the change in seasonal vegetation productivity (NDVI) in their surrounding area, assessed the composition of diet and microbiome, assessed the drivers and extent of seasonal shifts, and finally connected seasonal dietary shifts to indicators of female breeding success.

### Environmental variation

For black rhino (linear mixed effects model: standard deviation of random effect = 0.015, AIC = −1098.91) NDVI increased from pre-rain to post-rain ($\beta = 0.073$, se = 0.0021, $t = 34.62$, df = 150.8, $p < 0.001$) was lowest on Ol Jogi and highest on Ol Pejeta (Lewa to Ol Jogi: $\beta = -0.022$, se = 0.0033, $t = -6.67$, df = 60.7, $p < 0.001$, Lewa to Ol Pejeta: $\beta = 0.030$, se = 0.0038, $t = 8.00$, df = 69.3, $p < 0.001$, Fig. 2). For Grevy's zebra (linear regression: $R^2 = 0.80$, df=150), NDVI also increased pre-rain to post-rain ($\beta = 0.029$, se = 0.0028, $t = 10.02$, $p < 0.001$), although there was not much seasonal chance on Westgate. NDVI was lowest by far on Westgate and highest on Lewa (Lewa to Mpala-Ol Jogi: $\beta = -0.013$, se = 0.0034, $t = -3.80$, $p < 0.001$, Lewa to Westgate: $\beta = -0.078$, se = 0.0038, $t = -20.66$, $p < 0.001$, Fig. 2).

### Diet

The three plant families with the highest mean relative read abundances (which we will refer to as relative abundances) across reserves and seasons for black rhino were Fabaceae (woody plants and legumes; mean of 45%), Ebenaceae (evergreen trees and shrubs; mean of 23%) and Poaceae (grasses; mean of 10%). Grevy's zebra diets were made up of two major plant families across reserves and seasons; Poaceae (mean of 67%) and Fabaceae (mean of 29%). The highest utilised genera of Fabaceae and Poaceae differed between the two species. Within Fabaceae, black rhino consumed a high proportion of acacia (*Vachellia* and *Senegalia*) whereas zebra primarily optimised *Indigofera*. Among Poaceae, *Cenchrus* was the only genus that formed a high proportion of black rhino diets, whereas Grevy's zebra ate a lot of both *Cenchrus* and *Digitaria*.

Black rhino and Grevy's zebra diet composition varied across reserves and sampling seasons (Fig. 3; Supplementary Data 1&2, Supplementary Table 1). These differences were explained by variations in NDVI and rainfall (Table 1). Black rhino diet was most dispersed on Lewa, and least dispersed on Ol Pejeta, while Grevy's zebra diet was least dispersed on Westgate, but similarly dispersed on the other two reserves (Supplementary Fig. 1 and Supplementary Table 2).

In black rhino, the first principal component of diet composition explained 13.0% of the variation and was positively correlated with rainfall ($\beta = 0.011$, se = 0.0019, $t = 5.51$, df = 212, $R^2 = 0.12$, $p < 0.001$) and NDVI ($\beta = 0.018$, se = 0.0057, $t = 3.23$, df = 212, $R^2 = 0.04$, $p = 0.0014$). PC2 explained 9.5% of variation and was negatively correlated with rainfall ($\beta = -0.012$, se = 0.0019, $t = -6.52$, df = 212, $R^2 = 0.16$, $p < 0.001$) and NDVI ($\beta = -0.049$, se = 0.0048, $t = -10.11$, df = 212, $R^2 = 0.32$, $p < 0.001$, Fig. 3a). In Grevy's zebra, the first principal component of diet composition explained 23.3% of variation and was positively correlated with rainfall ($\beta = 0.0099$, se = 0.0008, $t = 12.06$, df = 152, $R^2 = 0.49$, $p < 0.001$) and NDVI ($\beta = 0.066$, se = 0.0046, $t = 14.29$, df = 152, $R^2 = 0.57$, $p < 0.001$). PC2 explained 7.7% and was negatively correlated with rainfall ($\beta = 0.0032$, se = 0.0011, $t = 2.86$, df = 152, $R^2 = 0.045$, $p = 0.0048$) but was not correlated with NDVI (Fig. 3c).

Across reserves within both species, the proportion of Poaceae (grasses) in the diet increased with increasing NDVI but saturated at different proportions (Table 2, Fig. 4). Across all reserves, the proportion of Fabaceae (legumes and woody plants) in the diet of black rhino and Grevy's zebra decreased with increasing vegetation productivity (Table 2 and Fig. 4). The proportion of Ebenaceae (evergreen trees and shrubs) in the black rhino diets was not associated with vegetation productivity (Table 2).

In black rhino, dietary breadth (dietary Shannon-Weaver diversity index) increased linearly with NDVI. In Grevy's zebra dietary breadth followed a negative quadratic with increasing NDVI (Table 3 and Fig. 5). Black

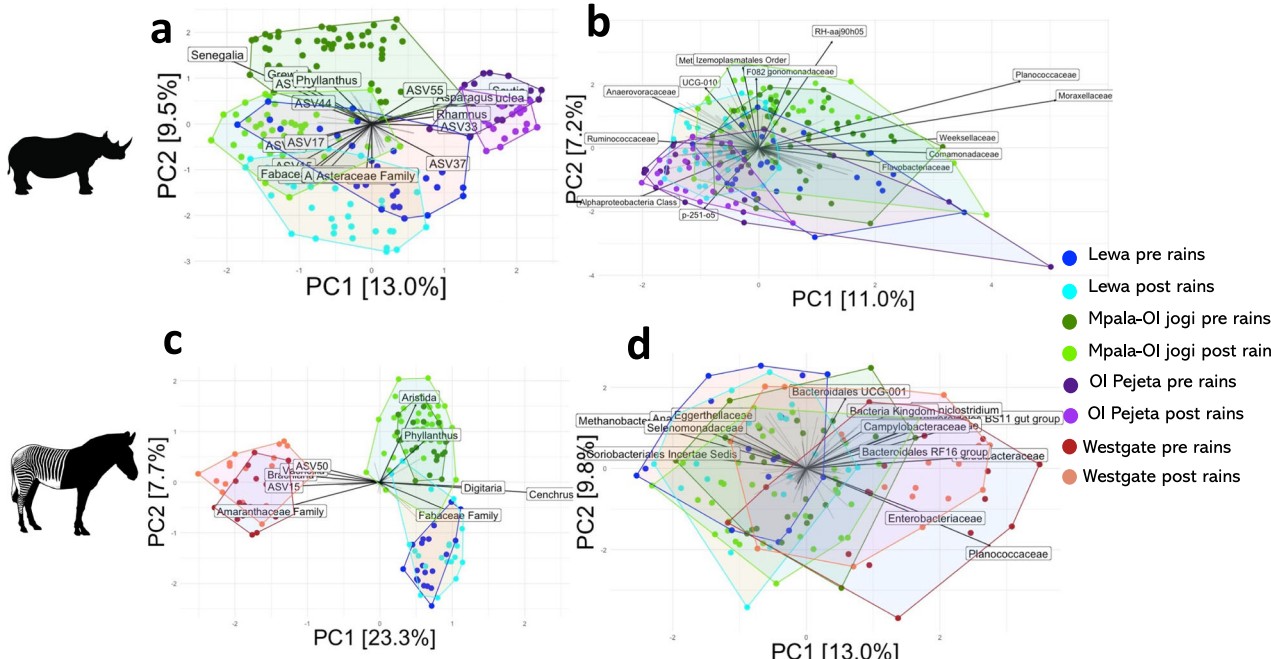

**Fig. 3 | Environmental variables drive differences in black rhino and Grevy's zebra dietary and microbiome composition across an ecological gradient.** PCA ordinations of dietary (black rhino: **a**, $n = 214$; Grevy's zebra: **c**, $n = 154$) and microbiome (black rhino: **b**, $n = 199$; Grevy's zebra: **d**, $n = 155$) beta diversity, CLR-transformed and agglomerated at the genus level for diet and family level for the microbiome. Arrows indicate loadings of genera with the length of the arrow indicating the magnitude of the loading score. *Eriochloa*, *Digitaria*, *Aristida* and *Cenchrus* are grasses. *Indigofera* and *Vachellia* are legumes. Amaranthaceae family is depicted as ASVs that could not be assigned to the genus. Some genera which loaded (>0.4 or < − 0.4) are not displayed for clarity. Dark blue represents samples taken on Lewa pre-rains, and light blue post-rains. Dark green represents samples taken on Mpala or Ol Jogi pre-rains, and light green post-rains. Dark purple represents samples taken on Ol Pejeta pre-rains, and light purple post-rains. Dark red represents samples taken on Westgate pre-rains and orange post-rains.

**Table 1 | perMANOVA with 10,000 permutations for weighted Unifrac dissimilarity in diet and microbiome composition across ecological gradient varying with environmental variables and dietary variables for microbiome**

| Diet or microbiome | Species | | Variable | Df (explanatory variable, total) | SS | $R^2$ | Pseudo-F | $p$ |
|---|---|---|---|---|---|---|---|---|
| Diet | Black rhino | | Rainfall | 1, 213 | 0.035 | 0.10 | 25.3 | ≤0.0001*** |
| | | | NDVI | 1, 213 | 0.014 | 0.040 | 9.89 | ≤0.0001*** |
| | Grevy's zebra | | Rainfall | 1, 153 | 0.18 | 0.52 | 174.55 | ≤0.0001*** |
| | | | NDVI | 1, 153 | 0.014 | 0.041 | 14.00 | 0.0002*** |
| Microbiome | Black rhino | Model 1 | Population | 2, 198 | 0.32 | 0.098 | 10.67 | ≤0.0001*** |
| | | Model 2 | NDVI | 1, 191 | 0.072 | 0.023 | 4.60 | ≤0.0001*** |
| | | | Poaceae (grasses) | 1, 191 | 0.031 | 0.0099 | 2.00 | 0.0266* |
| | | | Fabaeceae (legumes) | 1, 191 | 0.072 | 0.023 | 4.62 | ≤0.0001*** |
| | | | Ebenaceae | 1, 191 | 0.078 | 0.025 | 5.01 | 0.0002*** |
| | Grevy's zebra | Model 1 | Population | 2, 154 | 0.21 | 0.14 | 6.24 | ≤0.0001*** |
| | | Model 2 | NDVI | 1, 148 | 0.088 | 0.063 | 9.96 | ≤0.0001*** |
| | | | Poaceae (grasses) | 1, 148 | 0.019 | 0.014 | 2.23 | 0.051 |
| | | | Fabaceae (legumes) | 1, 148 | 0.0066 | 0.0048 | 0.74 | 0.56 |

NDVI denotes normalised difference vegetation index as a proxy of grass availability and forage. Variable = predictors in the model, Df = degrees of freedom, SS = sum of squares, $R^2$ = percentage of variance explained by the predictor, Pseudo-F = pseudo-$F$-value, $p$ = $p$-value with significance codes *** = $p < 0.001$, ** = $00.1 < p < 0.01$, * = $0.01 < p < 0.05$.

rhino dietary breadth followed a negative quadratic with increasing proportion of Poaceae, Fabaceae and Ebenaceae in the diet, such that dietary diversity was highest when individuals consumed intermediate amounts of each of these families (Table 3 and Fig. 5). In Grevy's zebra, dietary breadth followed a negative quadratic with increasing proportion of Poaceae in the diet (Table 3 and Fig. 5).

For both species, there appeared to be a trade-off between grass and legumes (Fig. 6). In black rhino, the proportion of Poaceae in the diet decreased with increasing Fabaceae (Spearman's rank correlation with Holm correction: $r = -0.30$, 95% CI = lower = −0.42, upper = −0.17, $t = -4.59$, df = 212, $p < 0.001$, Fig. 6) and the proportion of Fabaceae decreased with increasing Ebenaceae ($r = -0.56$, 95% CI = lower = −0.64, upper = −0.46, $t = -9.76$, df = 212, $p < 0.001$, Fig. 6), but there was no correlation between Poaceae and Ebenaceae. At a reserve level, the correlation between Fabaceae and Poaceae was only significant on Ol Pejeta. (Supplementary Table 3). Grevy's zebra switched to a legume (Fabaceae) based diet when they could not

**Table 2 | Outputs of models used to test whether the relative abundance of each plant family was significantly correlated with NDVI**

| Species | Variable | Standard deviation of random effect | AIC | Df (explanatory variable, total) | Variable | β | se | t | p |
|---|---|---|---|---|---|---|---|---|---|
| Black rhino | Proportion of Poaceae (grasses) in the diet | 0.043 | −380.33 | 1, 174.3 | NDVI | 0.0028 | 0.00043 | 6.45 | <0.001*** |
| | Proportion of Fabaceae (legumes) in the diet | 0.12 | 9.96 | 1, 183.72 | NDVI | −0.0037 | 0.0011 | −3.43 | <0.001*** |
| | Proportion of Ebenaceae in the diet | 0.17 | −51.36 | 1, 155.2 | NDVI | −0.0011 | 0.00085 | −1.25 | 0.21 |
| Grevy's zebra | Proportion of Poaceae (grasses) in the diet | | −76.54 | 1, 153 | NDVI | 2.88 | 0.19 | 15.52 | <0.001*** |
| | | | | 1, 153 | $NDVI^2$ | −1.13 | 0.19 | −6.08 | <0.001*** |
| | Proportion of Fabaceae (legumes) in the diet | | −85.23 | 1, 153 | NDVI | −2.58 | 0.18 | −14.30 | <0.001** |
| | | | | 1, 153 | $NDVI^2$ | 0.92 | 0.18 | 5.08 | <0.001*** |

There are two linear regression models for Grevy's zebra, with Poaceae and Fabaceae relative abundances as dependent variables, and three linear mixed effect models for black rhino, using black rhino ID as a random effect, with Poaceae, Fabaceae and Ebenaceae relative abundances as dependent variables. All predictors were included as second-order polynomial predictors, which were dropped if they were not significant. Variable = predictors in the model with a superscript 2 indicating predictors included as second-order polynomials. Df = degrees of freedom, β = regression coefficient, se = standard error, t = t-value, p = p-value with significance codes *** = p < 0.001, ** = 0.001 < p < 0.01, * = 0.01 < p < 0.05.

saturate their diet with grass. Individuals in Lewa and Mpala-Ol Jogi maintained the highest level of grass consumption across seasons. Individuals in Westgate did not display a period during our study where the average diet was majority grass (>50%). In Grevy's zebra across reserves the proportion of Poaceae increased with decreasing Fabaceae (Spearman's rank correlation with Holm correction: $r = -0.93$, 95% CI = lower = −0.97, upper = −0.87, df = 152, $p < 0.001$, Fig. 6). *Indigofera spp* were the major alternative food type consumed by Grevy's zebra across reserves and seasons ($r = -0.91$, 95% CI = lower = −0.95, upper −0.85, df = 152, $p < 0.001$).

**Microbiome**

Grevy's zebra microbiomes are dominated by Lachnospiraceae, which have a slightly higher relative abundance on Lewa and Mpala-Ol Jogi of 20-25% than Westgate. No bacterial taxa are as prevalent in the black rhino microbiome, with the order WCHB-41 having the highest average relative abundance of around 15%, although this is highest on Ol Pejeta post-rains with an average of 21%. This order is the second most common on average in zebra. The most common bacterial taxa are similar between both herbivore species, although Christensenellaceae and Oscillospiraceae are more common in Grevy's zebra, whereas Spirochaetaceae and Moraxellaceae are more common in black rhino. There are no clear seasonal changes in the relative abundance of the most common bacterial taxa across reserves, except for Spirochaetaceae and Moraxellaceae in black rhino which both decrease post-rains across all three reserves. Microbiome composition varied across reserves in both species (Table 1 and Supplementary Data 3 & 4). In black rhino, microbiome composition was influenced by the proportion of Poaceae, Fabaceae and Ebenaceae in the diet as well as NDVI (Table 1). In Grevy's zebra, microbiome composition was influenced by just NDVI (Table 1).

Individuals with more similar diets had more similar microbiomes in both species (black rhino – Euclidean distance of CLR-transformed abundances Mantel statistic $r = 0.16$, $p = 0.002$, $n = 183$—Weighted UniFrac Mantel statistic $r = 0.16$, $p = 0.008$, n = 183, Grevy's zebra, Euclidean distance of CLR-transformed abundances Mantel statistic $r = 0.13$, $p = 0.001$, $n = 152$—Weighted Unifrac Mantel statistic $r = 0.20$, $p = 0.001$, $n = 153$). In black rhino, individual seasonal dietary shift was not correlated with microbiome shift ($\beta = 0.20$, se = 0.19, $t = 1.02$, df = 51, $R^2 = 0.00093$, $p = $ NS).

In black rhino, the first principal component of microbiome composition explained 11.0% of variation and negatively correlated with NDVI ($\beta = -0.03$, se = 0.005, $t = -5.71$, df = 197, $R^2 = 0.14$, $p < 0.001$) and proportion of Ebenaceae ($\beta = -1.09$, se = 0.31, $t = -3.46$, df = 190, $R^2 = 0.054$, $p = <0.001$), and positively correlated with proportion of Fabaceae ($\beta = 0.88$, se = 0.27, $t = 3.22$, df = 190, $R^2 = 0.046$, $p = 0.001$). Bacterial families Ruminococcaceae, Anaerovoracaceae, Anaerovoracaceae, Desulfovibrionaceae, Acidaminococcaceae (loading scores > 0.7) negatively loaded onto PC1 while Xanthomonadaceae, Flavobacteriaceae, Comamonadaceae, Weeksellaceae, Planococcaceae positively loaded onto PC1 (Supplementary Data 5). PC2 explained 7.2% of variation was positively correlated with Fabaceae ($\beta = 1.25$, se = 0.27, $t = 4.71$, df = 190, $R^2 = 0.10$, $p < 0.001$, Fig. 3b). Furthermore, rh-aaj90h05, Izemoplasmatales, Methanocorpusculaceae, Dysgonomonadaceae, f082, Planococcaceae, Mycoplasmataceae, ucg-010 and Anaerovoracaceae positively loaded onto PC2 (loading scores > 1.0) (Supplementary Data 6).

In Grevy's zebra, the first principal component of microbiome composition explained 13.0% of variation and negatively correlated with NDVI ($\beta = -0.057$, se = 0.007, $t = -7.96$, df = 153, $R^2 = 0.28$, $p < 0.001$) and the proportion of Poaceae in the diet ($\beta = -2.31$, se = 0.28, $t = -8.23$, df = 147, $R^2 = 0.31$, $p < 0.001$), and positively correlated with proportion of Fabaceae ($\beta = 2.38$, se = 0.30, $t = 7.97$, df = 147, $R^2 = 0.30$, $p < 0.001$, Fig. 3d). Bacterial families including Planococcaceae, Paludibacteraceae, Bacteroidales bs11 gut group, Ruminiclostridium, Enterobacteriaceae, Desulfovibrionaceae Bacteroidales rf16 gut group and Campylobacteraceae negatively loaded onto PC1 (Loading scores >1) and are therefore associated with high NDVI and grass in the diet (Supplementary Data 7). PC2 explained 9.8% of the variation but was not associated with NDVI, Poaceae or Fabaceae (Supplementary Data 8).

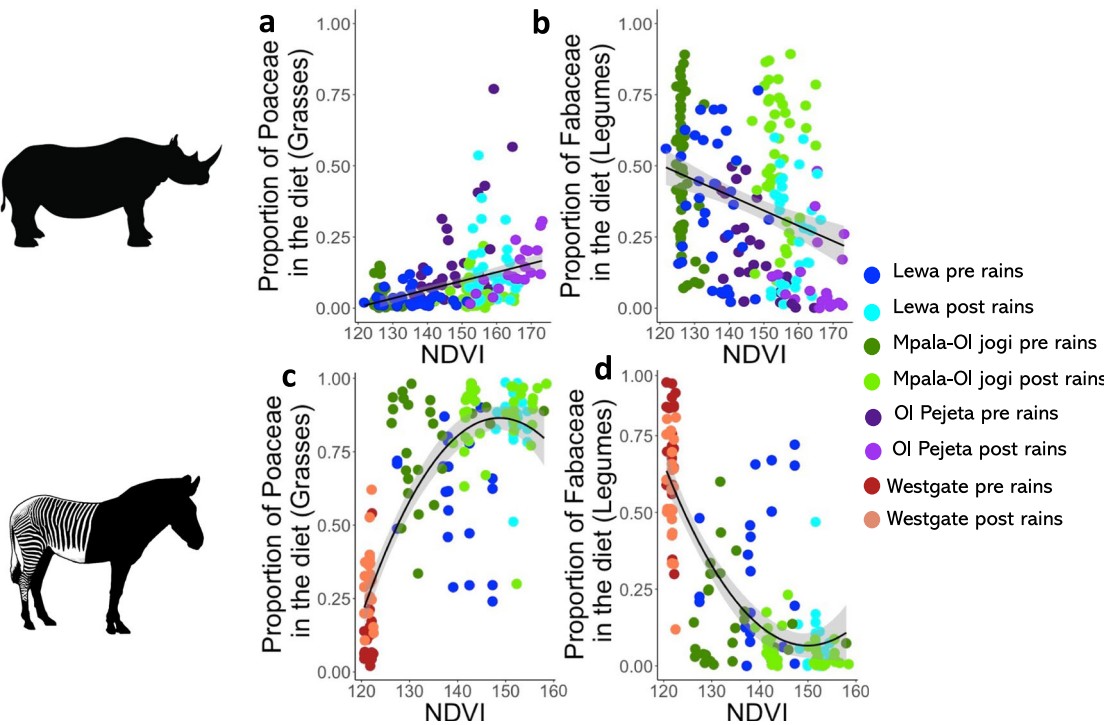

**Fig. 4 | Diet composition in black rhino and Grevy's zebra is associated with vegetation productivity.** The proportion of Poaceae in the diet increased with increasing NDVI in both black rhino (**a**, $n = 214$) and zebra (**c**, $n = 154$), whereas the proportion of Fabaceae decreased with increasing NDVI (**b**, **d**). Black lines represent the least square line of best fit, and grey areas represent the 95% confidence interval.

Dark blue represents samples taken on Lewa pre-rains, and light blue post-rains. Dark green represents samples taken on Mpala or Ol Jogi pre-rains, and light green post-rains. Dark purple represents samples taken on Ol Pejeta pre-rains, and light purple post-rains. Dark red represents samples taken on Westgate pre-rains, and orange post-rains.

**Table 3 | Outputs of models used to test whether dietary breadth (Shannon-Weaver diversity index) significantly correlated with NDVI and plant family relative abundances**

| Model | Species | Standard deviation of random effect | AIC | Variable | Df (explanatory variable, total) | β | se | t | p |
|---|---|---|---|---|---|---|---|---|---|
| Dietary breadth vs. NDVI | Black rhino | 0.18 | 281.14 | NDVI | 1, 194.4 | 0.0068 | 0.0021 | 3.32 | 0.011** |
| | Grevy's Zebra | | 196.28 | NDVI | 1, 153 | 1.29 | 0.45 | 2.878 | 0.004** |
| | | | | NDVI$^2$ | 1, 153 | −1.55 | 0.45 | −3.445 | <0.001*** |
| Dietary breadth vs key diet items | Black rhino | 0.034 | 84.81 | Fabaceae (legumes) | 1, 183.77 | −4.09 | 0.41 | −9.86 | <0.001*** |
| | | | | Fabaceae (legumes)$^2$ | 1, 206.01 | −1.64 | 0.31 | −5.32 | <0.001*** |
| | | | | Poaceae (grasses) | 1, 193.72 | 0.34 | 0.33 | 1.03 | 0.31 |
| | | | | Poaceae (grasses)$^2$ | 1, 204.75 | −0.87 | 0.29 | −2.99 | 0.0031** |
| | | | | Ebenaceae | 1, 144.60 | −4.95 | 0.38 | −12.78 | <0.001*** |
| | | | | Ebenaceae$^2$ | 1, 197.30 | −1.46 | 0.29 | −4.99 | <0.001*** |
| | Grevy's zebra | | 150.89 | Poaceae (grasses) | 1, 153 | 2.02 | 0.39 | 5.19 | <0.001*** |
| | | | | Poaceae (grasses)$^2$ | 1, 153 | −2.80 | 0.39 | −7.19 | <0.001*** |

All models have dietary breadth as the dependent variable. The results of two linear regression models for Grevy's zebra and two linear mixed effect models for black rhino, using black rhino ID as a random effect, are presented. All predictors were included as second-order polynomial predictors, which were dropped if they were not significant. Variable = predictors in the model with a superscript 2 indicating predictors included as second-order polynomials, Df = degrees of freedom, β = regression coefficient, se = standard error, t = t-value, p = p-value with significance codes *** = $p < 0.001$, ** = $0.001 < p < 0.01$, * = $0.01 < p < 0.05$.

### Fitness proxies

Black rhino female inter-calving interval increased with higher dietary shifts, such that individuals with larger dietary shifts bred more slowly (GLM with Gamma distribution: Null deviance 2.05 on 23 df, Residual deviance 1.65 on 22 df, $β = −0.072$, se = 0.031, $t = −2.34$, $p = 0.029$). There was a non-significant relationship between the proportion of legumes in the diet pre- and post-rainfall and the percentage of foals in each Grevy's zebra reserve (Linear model: $β = 6.09$, se. = 2.73, $t = 2.22$, df = 6, $R^2 = 0.36$, $p = 0.068$),

however, this analysis is based on very few counties and a single time point (Fig. 7).

### Discussion

We used metabarcoding to identify changes in diet and microbiome composition between two periods with different vegetation productivity in black rhino and Grevy's zebra across an ecological gradient. Optimal foraging models predict that diet diversity should increase with resource scarcity as

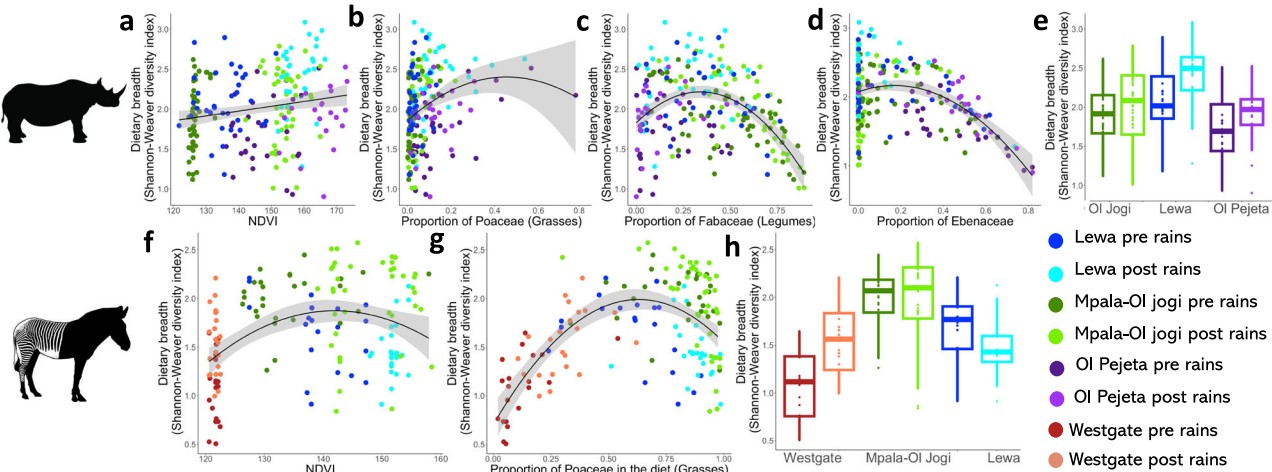

**Fig. 5 | Diet diversity dynamics of black rhino and Grevy's zebra across gradients of vegetation productivity and different relative abundances of important dietary plant families.** Black rhino dietary breadth ($n = 214$) (Shannon-Weaver diversity index) increased linearly with NDVI (**a**) and Grevy's zebra dietary breadth ($n = 154$) increased slightly but also followed a negative quadratic with NDVI (**f**). Dietary breadth varied with relative abundances of dietary plants. Black rhino dietary breadth followed a negative quadratic with the proportion of Poaceae (**b**) in the diet and decreased and followed a negative quadratic with proportion of Fabaceae (**c**) and Ebenaceae (**d**) in the diet. **e** Box plot of black rhino dietary diversity across reserves. Grevy's zebra dietary breadth increased, and followed a negative quadratic with proportion of Poaceae in the diet (**g**). **h** Box plot of zebra dietary diversity across reserves. Black lines represent least square line of best fit, and grey areas represents the 95% confidence interval. Dark blue represents samples taken on Lewa pre-rains, and light blue post-rains. Dark green represents samples taken on Mpala or Ol Jogi pre-rains, and light green post-rains. Dark purple represents samples taken on Ol Pejeta pre-rains, and light purple post-rains. Dark red represents samples taken on Westgate pre-rains, and orange post-rains.

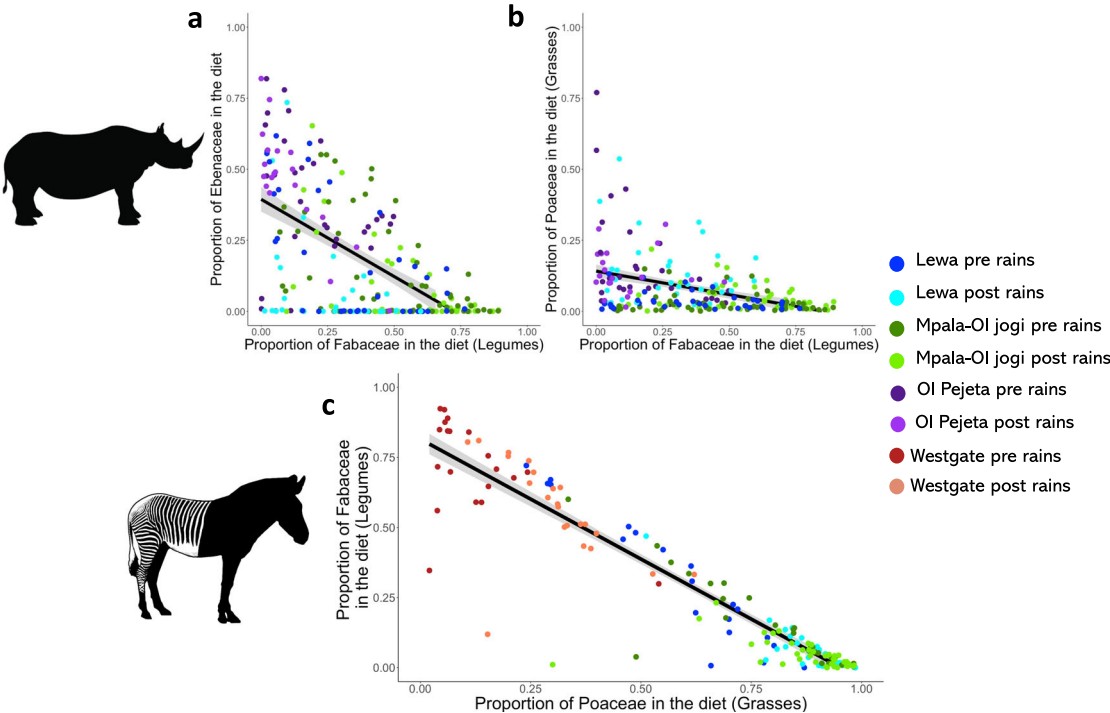

**Fig. 6 | Diet switching in black rhino and Grevy's zebra.** Black rhino ($n = 214$) and Grevy's zebra ($n = 154$) incorporated alternative dietary taxa into their diet with seasonal turnover. Black rhino Ebenaceae vs Fabaceae (**a**), Poaceae vs Fabaceae (**b**), Grevy's zebra Poaceae vs. Fabaceae (**c**). Black lines represent the least square line of best fit, and grey areas represent the 95% confidence interval. Dark blue represents samples taken on Lewa pre-rains, and light blue post-rains. Dark green represents samples taken on Mpala or Ol Jogi pre-rains, and light green post-rains. Dark purple represents samples taken on Ol Pejeta pre-rains, and light purple post-rains. Dark red represents samples taken on Westgate pre-rains, and orange post-rains.

valuable foods become less abundant[19]. However, we found more nuanced relationships between dietary responses, foraging strategy and vegetation productivity gradients than predicted by OFT.

Grevy's zebra foraged according to our predictions for grazers (Fig. 1). They switched to grasses as vegetation productivity increased and, with few exceptions, their diets were dominated by Poaceae at higher NDVI and Fabaceae (*Indigofera*) at lower NDVI, suggesting that *Indigofera* is a fallback food. The strength of the correlation between Fabaceae and Poaceae demonstrates that *Indigofera* has much lower value for Grevy's zebra than grasses. Consistent with dietary switching, dietary breadth was low in low

**Article**

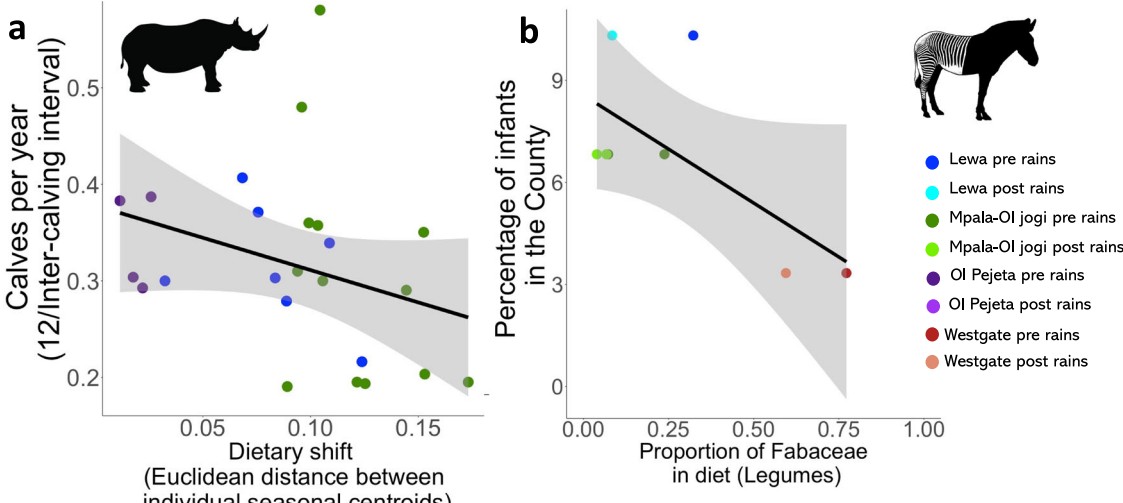

**Fig. 7 | Dietary switching is associated with female reproductive performance in black rhino, but there was no significant relationship between the mean relative abundance of Fabaceae in the diet and our measure of population performance in Grevy's zebra.** Black rhino calves per year (12/inter-calving interval (defined as age in months over 5 years/Number of calves)) decreased, indicating slower breeding, as dietary shift between pre-rain and post-rain sampling periods increases (**a**, $n = 24$). In Grevy's zebra, there was no significant relationship between the percentage of infants in the county, higher values of which indicate stronger population performance, and higher mean (averaged over pre- and post-rains) proportion of legumes in the zebras' diets in that county (**b**, $n = 8$). Black lines represent a least square line of best fit, and grey areas represent the 95% confidence interval. For black rhino, blue represents females on Lewa, green represents females on Jogi and purple represents females on Ol Pejeta. For Grevy's zebra, dark blue represents all individuals studied on Lewa pre-rains, and light blue post-rains. Dark green represents all individuals studied on Mpala pre-rains, and light green post-rains. Dark red represents all individuals studied on Westgate pre-rains, and orange post-rains.

NDVI contexts, where the diet was dominated by *Indigofera*, low in high NDVI contexts, where the diet was dominated by valuable grass species, and high in periods of intermediate vegetation, when individuals consumed a mix of legumes and limited grasses, resulting in a negative quadratic relationship between NDVI and diet diversity.

Black rhino dietary responses were not as straightforward, as they employed a mixed-feeder strategy year-round. It is likely that individuals employed different strategies, with some able to access valuable foods year-round, some switching seasonally, and others constrained to fallback foods by their ranges. This would explain why Fabaceae and Poaceae are negatively correlated for all individuals but not on Lewa and Ol Jogi, as more individuals on these reserves seemed to be constrained by lack of access to Poaceae even after the rains (Fig. 6). Higher NDVI was associated with a relative decrease in the consumption of Fabaceae, the family containing acacias, which are considered to be a preferred black rhino food[49,53]. The steep gradient of the negative relationship between vegetation productivity and Fabaceae consumption is difficult to explain if black rhino have a strong preference for acacias. The leaves of savanna trees increase in nutrient levels with increasing rainfall[54] and if Fabaceae species were optimal dietary taxa for black rhino, there are no clear expectations for individuals to supplement their diets with plants from other families as vegetation productivity increases. Black rhino may include other plant families in their diets to access certain nutrients or to avoid secondary compounds. Whilst acacia trees have been found to have relatively high crude protein content, their phenolic and tannin connection may reduce their digestibility[55]. Mineral content depends on soils, but acacias have been found to have high concentrations of many minerals including calcium, but low phosphorus[56].

This work suggests that black rhino have more flexible foraging strategies and a wider dietary niche than previously thought. This is supported by stable isotope ratio research that showed some black rhino in Laikipia have up to 40% of their diets made up of $C_4$/CAM plants for short periods, which is thought to occur after rains when grasses have the highest level of nutrition[48]. This calls into question conventional diet analyses which focus on leaf clipping to assess woody vegetation preferences[47], which cannot accurately assess grass use. Moreover, across the three study reserves, black rhino diets were spatially and individually variable (Supplementary Figs. 2–4). Several individuals also had diets that were comprised of more than 50% of plants in families other than Fabaceae, Poaceae and Ebenaceae. Theoretical models from other research suggest that the ability to change diets seasonally is advantageous for savanna herbivores and that mixed feeders have access to more and better food than specialised browsers or grazers[18]. Dietary flexibility may be a beneficial trait that allows black rhino to exploit the resources available across different habitats.

In Grevy's zebra, high vegetation productivity was associated with *Ruminiclostridium* and Campylobacteraceae including *Campylobacter spp*, *Escherichia-Shigella spp* and *Lysinibacillus spp* (Supplementary Data 4). *Ruminiclostridium* is crucial in maintaining the stability of the intestinal environment as they secrete short-chain fatty acids, which are conducive to maintaining the functionality and morphology of intestinal epithelial cells[57]. Some members of Campylobacteraceae including *Campylobacter spp*, *Escherichia-Shigella spp* and *Lysinibacillus spp*, are potential pathogens to humans and animals[58]. Zebra did not seem to conform to patterns predicted by the cost-of-switching hypothesis as average reserve level seasonal dietary shifts did not match microbiome shifts (Supplementary Table 4). The large shifts that were observed in the arid and grass-poor environment of Westgate suggest that one cost of reliance on fallback foods may be unstable microbiomes.

Our study does not support the cost of switching hypothesis for black rhino microbiomes either. However, in contrast to Grev's zebra, reliance on fallback foods did not seem to lead to microbiome instability as black rhino on Ol Jogi showed small shifts away from Fabaceae but had an average microbiome shift in between that of Ol Pejeta and Lewa (Supplementary Table 4). On Lewa and Ol Pejeta, microbiomes became much more dispersed before the rains and overlapped more with those of Ol Jogi black rhino, driven by shifts towards bacterial families including Planococcaceae, Moraxellaceae, Flavobacteriaceae, Weekselaceae and Dysgonomonadaceae (Supplementary Data 3).

It may be that seasonal shifts take place too quickly to affect microbiome for any given individual, and longer-term dietary and environmental differences between individuals drive microbiome variation at a population level. Whilst diet composition has been found to be a primary regulator of the microbial niche available in mammalian guts[59], microbiome composition is also driven by other factors. Secondary plant metabolites associated with certain taxa may affect particular bacterial taxa[60].

Our results have implications for the management of both species. Grass availability is a major factor determining habitat selection for Grevy's zebra[51]. Poor grass habitats likely represent marginal habitats for Grevy's zebra, as seen in the closely related Cape Mountain zebra (*Equus zebra zebra*)[28]. During the period of our study, Westgate appears to be a marginal habitat with an extended period of grass limitation, while Lewa and Mpala-Ol Jogi are higher quality, core habitats, where grass-rich diets can be maintained. Low availability of grass in some areas may be partly due to climate, but also due to domestic livestock. Cattle preferentially consume grass[61] and have been shown to compete with elephants in this region for grass[62]. Diet diversity decreased post-rains at Lewa, as diets were grass-rich. In Mpala-Ol Jogi diet diversity was higher in both periods compared to the other reserves. In Westgate, zebras increased dietary breadth by adding grasses to a legume-dominated diet during periods of higher rainfall. High reliance on a single alternative food source during resource-scarce periods could increase intraspecific and interspecific competition for resources[63].

When environments become dry and NDVI decreases, Grevy's zebra can migrate to find more suitable conditions[64] resulting in seasonal dispersal dynamics across the metapopulation. This could be within a reserve, as we found large differences in the proportion of grass consumed by individuals on the same reserve, particularly during the dry seasons on Lewa and Mpala-Ol Jogi (Supplementary Fig. 5) indicating variable access to grass across these reserves, or across larger distances. Historically, zebra were able to track grass availability, which allows individuals to buffer their diets. The Laikipia-Samburu landscape is under pressure of fragmentation from fencing[65] with conservancies in the south and the west being disconnected from those in the central and northeastern regions[64]. Limiting the ability of grazing species, including Grevy's zebra, to seasonally track resources is likely to result in greater reliance on fallback foods, more frequent and pronounced diet switching and decreased resilience of their metapopulations. Future research, particularly on Westgate, to see what proportion of the population migrates, what proportion stays in place, how this varies by age and sex, and how fitness varies between the strategies would be important for conservation.

For black rhino, slower breeding was associated with greater dietary switching and rhino on Ol Jogi both consumed a high proportion of Fabaceae and exhibited greater dietary switching. There was also a positive correlation between the proportion of Fabaceae in the diets of females who have bred at least once and the magnitude of their dietary shifts (Supplementary Fig. 6 and Supplementary Table 5), suggesting that these individuals switch away from Fabaceae when they can. Previous work has shown that female black rhino on Ol Jogi exhibit lower breeding rates and higher mortality rates than Lewa and Ol Pejeta, and the population is more susceptible to extinction than in the other two reserves[66] where Fabaceae is a smaller component of the diet. Although black rhino on Ol Jogi increased consumption of Poaceae following rains, the relative proportion was generally quite low compared with the other better-performing populations. Dietary switching may therefore be a response that occurs in poor, marginal habitats where individuals fallback to utilising a high proportion of Fabaceae in times of scarcity and incorporate high-value plants when they become available. Strong seasonal dietary shifts may impact long-term reproductive trends and such shifts likely indicate marginal habitat and diet for black rhino. Unlike more nomadic species, black rhino are relatively strongly tied to local resource abundance. The diversity of foraging strategies and decrease of Fabaceae consumption during periods of higher vegetation productivity suggest that measuring habitat quality needs to go beyond assessing access to trees like *V. drepanolobiium*[67] and that carrying capacity estimates should be re-evaluated.

This work has important implications for future research into herbivore diets, and indeed any work that attempts to identify ecological niches or attribute variation in population performance to particular habitat characteristics. The dietary responses we observe in black rhino and Grevy's zebra may be common responses to lean and plentiful periods in other mixed feeders and grazers, allowing them to adjust to changes in abiotic and biotic conditions[46], and could act as a proxy for habitat quality. Seasonal

dietary shifts in strict browsers are also likely to follow the predictions of OFT but this remains to be tested.

Species can be very flexible in the diets that they consume, and it should not be assumed that the diet a species is observed to be eating in a particular area is optimal[28]. Protected area placement has often been controlled by social and economic factors[68,69], rather than ecological ones such as optimal habitat for a target species. Even in reserves created for one flagship species, it cannot be guaranteed that the diet that the species is consuming there is optimal or near-optimal[70]. This work provides further evidence that savanna herbivores are facultative generalists that can eat a range of plants but specialise in certain taxa in any particular place and time[46], seasonal dietary strategies are a vital determinant of savanna herbivore population dynamics[18], and diet switching over time and space is one of the key species responses to ecological gradients across habitats. Together, these factors ultimately determine the realised niche, range dynamics and limits of these herbivores[71,72].

Foraging or other processes should not just be studied in one season or area, which will miss the dietary switching we have demonstrated to be important. This risks creating a species stereotype, which is a biased or false understanding of aspects of a species' ecology and false conclusions about its niche[73]. For example, if studies only take place in marginal habitats, where valuable foods are rarely or never available, then it may be assumed that fallback foods are more valuable than they are. This can have knock-on effects on conservation, including misestimation of carrying capacity and the placement of reserves in marginal habitats, which may undermine our ability to effectively protect and restore populations of endangered species.

## Methods

### Study sites

Faecal samples from black rhino and Grevy's zebra were collected at five reserves in Kenya in 2018 and 2019: Ol Pejeta Conservancy (black rhino), Ol Jogi Conservancy (black rhino and Grevy's zebra), Lewa Wildlife Conservancy (black rhino and Grevy's zebra), Mpala Research Centre (Grevy's zebra) and Westgate Community Conservancy (Grevy's zebra) (Supplementary Fig. 7). Plant communities in this ecosystem are dominated by woody species from the Fabaceae (including *Vachellia* and *Senegalia* species) and Ebenaceae (including *Euclea divornum)* families, and Poaceae (grasses). Ol Pejeta in Laikipia Country (0.02°N, 36.90°E) has an average annual rainfall of around 740 mm and the habitat cover types are dominated by grassland, *V. drepanolobium* wooded grassland and *Euclea divinorum* thicket[49,74]. Ol Jogi also in Laikipia County (0.32°N, 36.98°) has an average annual rainfall of around 570 mm and is dominated by *Vachellia* and *Senegalia* woodland/thicket and has a smaller proportion of *V. drepanolobium* wooded grassland than the other two reserves[49]. Lewa is sited in Meru County (0.20°N, 37.42°E), has an average annual rainfall of 570 mm and is dominated by *V. drepanolobium* wooded grassland with other habitats including mixed species bushland and mountain forest[49,75]. Mpala is adjacent to Ol Jogi in Laikipia County (0.31°N, 36.96°E) and has an annual rainfall of around 600 mm. It is dominated by *V. drepanalobium* bushland, *Senegalia brevispica* thicket and grassland[76]. As Mpala and Ol Jogi are adjacent properties in which Grevy's zebra, but not black rhino which are not present on Mpala, can move freely between they were analysed as one area. Westgate is in Samburu County (0.81°N, 37.3°E), has an average annual rainfall of around 190 mm and is savanna grassland with varying densities of *Vachellia, Commiphora* (woody shrubs and trees in the Burseraceae family)*, Boscia* (woody shrubs and trees in the Capparaceae family) *and Grewia* (woody shrubs and trees in the Malvaceae family)[77]. This region of Kenya traditionally has two annual rainy seasons; the long rains March-May and short rains October–December. Monthly rainfall peaks at around 100 mm in April and November on Lewa and Ol Pejeta, and around 80 mm on Mpala and Ol Jogi. Mpala, Ol Jogi and Ol Pejeta have rainfall relatively well-spread throughout the rest of the year, with around 20-50 mm per month and a small peak in August, while Lewa has a pronounced dry period July-September[49]. Westgate has a similar timing of rains but lower amounts at all times of year compared to the other reserves. It should be noted that

Kenya experienced a drought from 2020–2023, but our samples were collected before this disruption to rainfall patterns.

## Obtaining seasonal data on food availability

All geographical analyses were conducted in QGIS version 3.16 (QGIS.org 2020).

We estimated the amount of rainfall in the study reserves during field seasons using the Climate Hazards Group Infrared Precipitation combined with the Station observations (CHIRPS) dataset at a resolution of 0.05°[78,79]. As the rainfall data is at a relatively coarse resolution, we used the rainfall of the pizel under each sample. We used rainfall over the 30 days previous to sample collection for the pixel under each sample, which is roughly equivalent to 5.5 km², to give a biologically relevant measure of water availability[80].

In order to estimate vegetation productivity, we used the Normal Difference Vegetation Index (NDVI). This measure of habitat greenness correlates with forage biomass in East Africa[81] and has been used as a proxy of forage palatability and quality for Grevy's zebra[51]. Cleaned and processed 10-day composite eMODIS NDVI values that contained the days of sample collection for all sample seasons and sites were sourced from the US Geology Survey (USGS Famine Early Warning Systems Network)[82], as detailed below.

We extracted NDVI in areas we estimated were utilised by each individual during our sampling periods. Black rhino home ranges vary in size and are generally larger than the resolution of the NDVI data[50]. We were not interested in exact home ranges but rather a way to estimate the possible area used by individual black rhino across the sampling periods. We estimated this utilised area using the dung samples used in this study, and more which were collected using the same method but were not included in the metabarcoding analysis. These were collected over the two field seasons described below and another fieldwork season on the same sites between October–December 2017. This was a total of 449 samples from 128 individuals. The number of samples per individual ranged from 1 to 13. One hundred individuals had more than one sample, with a mean of 4.2 samples per individual and a mode of 3. The size of each individual's utilised area was estimated by creating a minimum bounding circle around each individual's samples, clipped at the boundary of the reserves. This gave a mean area of 15.5 km². The 28 individuals which had one sample were assigned a utilised area with a radius that was equivalent to the mean of the radii of the other individuals, around 2.2 km. The mean NDVI was then calculated within each of these areas of utilisation.

We did not have definite repeat samples for Grevy's zebra, so used a different method to estimate the utilised area. As gut retention time for Grevy's zebra is approximately 24-48 h and Grevy's zebra can move ~2.3 km per day[52], the mean NDVI was calculated within a circle with a radius of 2.3 km centred on the sample collection point.

## Demographic data

The three black rhino reserves record the dates of births, deaths, imports (including the ages of imported individuals) and exports as part of their regular monitoring. The ages at which females died and gave birth have been accurately recorded since the foundation of each sanctuary. The reserves supplied these records and we formatted them so we could calculate what we called a female inter-calving interval. This is the age of the female over the age of five years, calculated in months, divided by the number of calves produced. We chose five years as, generally, the earliest that black rhino females can calve is five years old[83], although some females have been recorded to calve between the ages of four and five[84]. Other studies have estimated breeding success using the age of first calving, and a measure of inter-calving interval calculated by dividing the number of months between a female's first calf and the present/death by the number of calves produced. Our measure of female inter-calving interval incorporates information from both of these measures. We could calculate this metric for 24 females which were then included in the analysis, seven from Lewa, 13 from Ol Jogi and four from Ol Pejeta. Paternity is difficult to assign without the use of genetic

techniques[85,86], as mating is not always observed and females may mate with several males during oestrus. Due to this uncertainty, we did not include data on male breeding in the analyses.

For Grevy's zebra, we used the percentage of infants in each of Laikipia, Meru and Samburu counties from Rubenstein et al.[87] as an estimate of female reproductive rates.

## Sampling

This study was approved by the University of Manchester's Committee for the ethical review of category D research (Ref: 0030). We have complied with all relevant ethical regulations for animal use. For both black rhino and Grevy's zebra, we use faecal samples taken from males and females older than weaning age.

Faecal samples were collected over two field seasons (Black rhino: June–July 2018 and January–March 2019; Grevy's: July–August 2018 and January–February 2019). These periods were chosen as they are after and before periods of expected rain respectively[49]. All sampling was non-invasive as defined by Pauli et al.[88].

For black rhino, due to security and monitoring activities, each individual is known by name and it is usually possible to estimate their location on a particular day. Samples included in the analysis were either collected after observing defecation or could be assigned to an individual with high confidence. Only samples less than six hours old were collected and most were collected in the early morning and late afternoon. We collected samples from at least two complete boluses per dung pile, from several areas of each bolus and avoiding the surface 1 cm depth.

For zebra, samples were collected from individuals who had been observed defecating. Several whole droppings were taken from each dung pile. We photographed and recorded the sex of sampled individuals.

We conducted processing within a maximum of six hours from defecation, but most often within 2–3 h. We placed samples in sealed unused plastic bags, removed excess air, homogenised the samples in their sample bags, and then Grevy's zebra samples were stored as native dung at −20°C, until DNA extraction in November 2019. For black rhino samples, we removed around 3 g from the homogenised dung using sterile implements and stored them in 8 ml of 100% ethanol during the field season up to a maximum of nine weeks. Storage in ethanol has been shown to be effective for the extraction of DNA up to six months after sample collection[89]. Comparison of sequencing data obtained from faecal samples immediately frozen to those stored in ethanol has shown that while diversity metrics can be altered by storage methods, biologically relevant individual microbiome identity was retained[90]. These tubes were then stored at −20°C until DNA extraction in November 2019.

## DNA metabarcoding of the 16S rRNA gene and chloroplast *trnL* genes

We used 226 samples for the black rhino metabarcoding analysis: 73 from Lewa, 53 from Ol Pejeta and 100 from Ol Jogi, from a total of 88 different individuals. We used 158 samples for the zebra metabarcoding analysis, 45 from Lewa, 43 from Westgate, 36 from Ol Jogi, 28 from Mpala and 6 from Karisia. Zebra could move between the adjacent Ol Jogi, Mpala and Karisia areas so individuals from these areas were treated as one population for analysis.

We extracted DNA using the QIAamp DNA Stool Mini Kit (Qiagen, UK) according to the manufacturer's protocol with the addition of incubation at 95°C for 30 min. Extraction was carried out with an extraction blank in a laboratory designed to conduct molecular analyses, with separate pre- and post-PCR rooms and different equipment for extraction, PCR (including a PCR cabinet sterilised after each batch) and post-PCR processing. Products were checked using a Qubit 4 Fluorometer using a Qubit™ dsDNA BR Assay Kit (Invitrogen, CA, USA) according to the manufacturer's instructions.

We analysed samples for bacterial and plant composition using amplicon sequencing. For bacteria, we used 16 S rRNA[37,91]. For plants, we used the P6 loop of the chloroplast *trn*L(UAA) region[20,52]. We dual-indexed

amplicon sequences with index primers to allow for DNA sequences to be assigned to their sample. The 5' end of each forward amplicon primer was tagged with one of 16 8-nt multiplex identification (MIID) tags, and the 5' end of each reverse amplicon primer was tagged with one of twenty-four 8-nt MIID tags. These indexes could be combined to give 384 different combinations which allowed for all PCR products to be uniquely identifiable after they were pooled for sequencing. Primer sequences can be found in the Supplementary Information (Supplementary Tables 6–8).

We amplified the bacterial DNA and indexed it for the 16S rRNA gene (v4 region) in one round of PCR using dual-indexed forward and reverse primers[91]. PCRs were run in 30 µl reactions using 5x HOT FIREPol Blend Master Mix (Solis BioDyne, Estonia), 2 µM primers, and 3 µl of sample DNA using thermocycling conditions of 95 °C for 15 min; 25 cycles of 95 °C for 20 s, 50 °C for 60 s, 72 °C for 60 s; and a final extension at 72 °C for 10 min.

We amplified the plant DNA in 25 µl PCRs according to Kartzinel et al.[52] for the *trnL*-P6 region using Platinum Green Hot Start PCR 2X Master Mix (Thermo Fisher, MA, USA). 0.2 µM each primer [trnL(UAA)g/ trnL(UAA)h], and 2 µl of sample DNA using thermocycling conditions of 95 °C for 5 min; 35 cycles of 95 °C for 30 s, 55 °C for 40 s, 72 °C for 60 s; and a final extension at 72 °C for 10 min. We cleaned PCR products using HighPrep PCR clean-up beads (MagBio, USA) according to the manufacturer's instructions. We used a second round of PCRs to add the indexes to the amplicon primers in 25 µl reactions using KAPA HiFI Ready mix (Kapa Biosystems, MilliporeSigma, MI, USA), 1 µM index primes and 2 µl cleaned PCR product using thermocycling conditions of 95 °C for 45 s; 7 cycles of 98 °C for 20 s, 63 °C for 20 s, 72 °C for 2 min.

The method used was the same for both groups from this point, except for differing library concentrations, percentage of PhiX spikes and the Miseq kits used. After amplification, we cleaned PCR products using HighPrep PCR clean-up beads (MagBio, USA) according to the manufacturer's instructions. We then quality-checked PCR products on an Agilent 2200 TapeStation with High Sensitivity D1000 ScreenTape (Agilent Technologies, CA, USA) according to the manufacturer's instructions and then cleaned them using HighPrep PCR clean-up beads.

Sequencing bias is reduced by having roughly equal concentrations of DNA from each sample in the sequencing pool for the full sequencing run, which produces the analysable data. We used a titration sequencing run to estimate the amount of tagged DNA sequences present in each sample. We used 1 µl of product from each sample to create a titration pool. We determined the average fragment size using TapeStation, quantified the concentration on the Qubit, and used these to dilute the pool to 4 nM. We conducted a titration sequencing run using paired-end reads (2 × 150 bp) with a 50-cycle reagent kit (MiSeq Reagent Kit v2 MS-102-2001) at a concentration of 4pM for both libraries, with a 5% spike of PhiX Control v3 (Illumina, FC-110-3001) for the bacteria and 15% spike for diet on the Illumina MiSeq platform at the International Livestock Research Institute, Nairobi.

The results of the titration sequencing were used to create the pool for the full sequencing runs which produce the analysable data. the amount of each product included in the pool for the final sequencing run was inversely proportional to the occurrence of their tagged sequences in the titration run. This equalises the amount of DNA from each sample as far as possible.

For the full sequencing runs, we determined the average fragment size using TapeStation, quantified the concentration on the Qubit, and used these to dilute the pool to 4 nM. We conducted diet sequencing using paired-end reads (2 × 250 bp) with 500-cycle reagent kits (MiSeq Reagent Kit v2 MS-102-2003) and a 15% spike of PhiX Control v3 at a library concentration of 4pM. We conducted bacteria sequencing using paired-end reads (2 × 300 bp) with a 600-cycle reagent kit (MiSeq Reagent Kit v3 MS-102-3003) at a concentration of 12pM and a 5% spike of PhiX Control v3 (Illumina, FC-110-3001). All sequencing runs were carried out on the Illumina MiSeq platform at the International Livestock Research Institute, Nairobi.

**Bioinformatic processing of DNA sequences**. The target amplicon for the diet analysis was shorter than the paired reads which were sequenced, so adapters were present in the resulting sequences. We used Cutadapt 2.1[92] to remove the forward and reverse adapters present in the data. One of the black rhino samples had no reads and could not be put through the Cutadapt pipeline, and was therefore excluded from the remainder of the analysis. All remaining processing and analysis were conducted in Rstudio[93] for R (v4.2.0)[94]. We processed resulting amplicon sequences in DADA2 (v1.18.0)[95], using the package to filter, trim and denoise, merge paired reads and remove chimeras. Reference libraries are available for rRNA bacterial studies and we assigned taxonomy using the SILVA v138 database[96,97]. We assigned plant taxonomy using the P6 loop of the chloroplast *trnL*(UAA) region, utilising version 2.0 of a library constructed at Mpala Research Centre, one of the zebra study sites[98,99]. We used these to assign each unique sequence (as amplicon sequence variants; ASV) to a known taxon if possible. DADA2 is not able to assign taxonomy to sequences under 50 bases long. This is only relevant for the plant analyses, as 122 reference sequences in the library are shorter than 50 bases. Among these reference sequences, five belonged to Fabaceae, four to Poaceae, and none to Ebenaceae. ASVs that exactly matched the sequences in these three families were assigned manually. All five Fabaceae, and two of the Poaceae sequences, were present in the dataset.

A full breakdown of the numbers of sequences and ASVs at each processing step, and the proportion of final ASVs assigned to family and genus taxonomic levels, is presented in the Supplementary Information (Supplementary Tables 9–18). We considered that a sample having under 1000 total reads for diet and under 2000 for bacteria after processing was indicative of sequencing failure, and removed those samples from further analysis. We removed 12 samples for the black rhino diet, four samples for the zebra diet, 27 samples for the black rhino microbiome and three samples for the zebra microbiome. This left us with the following sample sizes. Rhino diet: Lewa-67, Ol Jogi-97, Ol Pejeta-50. Rhino microbiome: Lewa-59, Ol Jogi-92, Ol Pejeta-48. Zebra diet: Lewa-45, Westgate-42: Mpala-Ol Jogi (also includes Karisia samples)−67. Zebra microbiome: Lewa-45, Westgate-43: Mpala-Ol Jogi (also includes Karisia samples)−67.

Further analysis was conducted using the phyloseq package[100] which combines the final ASV table, taxonomy table and sample metadata. Analyses were used to test what environmental factors affected diet and microbiome composition, what plants were favoured during times of abundance, whether important plant families were substituted for each other seasonally, and whether seasonal diet shifts drove changes in the microbiome. We also tested whether the proportion of grass in the zebra diets and the size of seasonal dietary shifts in black rhino affected demographic indicators of fitness.

We converted the number of reads to relative read abundance (which call relative abundance)[101–103]. Analyses used relative abundances at the genus and family levels. The relative abundance of the top 20 most abundant genera of plants and the 20 most abundant bacterial families, across reserves and seasons are presented in Supplementary Data 1–4, but further analysis was carried out with all ASVs. For individual black rhino sampled more than once in one sampling season, there were no consistent patterns of variation in dietary composition between early and late in each sampling season (Supplementary Figs. 8 & 9), so we used the mean relative abundances.

**Statistics and reproducibility**

We describe the statistical tests we used for each individual analysis carried out in R. These are similar for the two species, but we used mixed effects models with individual ID as a random effect for black rhino and not for Grevy's zebra, because we had information on the individual that each sample was collected from for black rhino. Sample sizes are generally controlled by the number of samples that were left after bioinformatic processing (214 and 199 for the black rhino diet and microbiome respectively, and 154 and 155 for Grevy's zebra diet and microbiome respectively). The sample size for each figure is presented in the legends, and degrees of freedom are presented with the results for each statistical test.

Whilst is difficult to apply strict principles of reproducibility to field-based ecological research, the sampling strategy was designed to give robust measures of differences across environmental gradients by sampling a range of different individuals, both male and female, right across each reserve and both pre- and post-rains. Grevy's zebra were not identified individually so there are no replicates for that species, but 38 black rhino individuals were sampled twice in the pre-rain period, and 29 post-rain. These repeat samples of the same black rhino individuals in the same sampling period show broadly similar dietary composition of the three study plant families although there is some variation. The diet and microbiome composition of study animals will vary naturally accordingly due to many factors, and so it is expected that sampling the same animal twice on different days will give different results even during the same season. Whilst there is intra-individual variation on each reserve, the clustering of each reserve on the principal component analyses (PCAs) and the consistent patterns of dietary and microbiome variation across these gradients and across many different individuals demonstrate reproducible effects of rainfall and NDVI.

## Vegetation productivity analyses

We analysed whether NDVI varied between reserve and sampling season at the utilised area level separately for each species. We used linear regression for the zebra, and linear mixed effects models, using the *lmer* function in the lme4 package, with black rhino ID as a random effect[104] for the black rhino. For both models, we log-transformed NDVI as the response variable because NDVI was right skewed.

## Diet analyses

Savanna herbivore diets can be described using the proportions of Fabaceae, Poaceae and all other families[46]. After examining the most prevalent plant families in the diet, we used the relative abundance of Fabaceae, Poaceae and Ebenaceae to test diet switching, and whether their presence in the diet was affected by environmental changes in rainfall and NDVI. We only included Fabaceae and Poaceae in the zebra analyses as they made up almost the entirety of their diets. We used linear regression for Grevy's zebra, and linear mixed effect models for black rhino using black rhino ID as a random effect, to test whether the relative abundance of each family was significantly correlated with NDVI. We ran all models with second-order polynomial predictors which were dropped if they were not significant. For these and all other linear and linear mixed effects models, we present AIC values which were calculated as part of the *glm* function for Grevy's zebra and using the *AIC* function for black rhino.

We calculated the Shannon-Weaver index of dietary alpha diversity using the *estimate_richness* function in the phyloseq package[100]. We then tested whether the alpha diversity was significantly predicted by NDVI and the relative abundance of each relevant plant family. We used linear regression for the zebra, and linear mixed effects models, with black rhino ID as a random effect, for the black rhino.

We used Spearman's rank correlation for the zebra to test whether there were significant relationships between the relative abundances of the relevant plant families and also Poaceae and *Indigofera*. Using the lava package[105] we calculated multilevel Spearman's correlations, controlling for black rhino ID after merging replicate samples within the season, to test whether there were correlations between the relative abundances of each of the three important dietary plant families. Correlations for all individuals are presented in the main text, and in Supplementary Table 3 for each individual black rhino reserve Holm corrections were used separately for the black rhino and zebra tests to control for multiple comparisons using the *p.adjust* function.

## Beta diversity analysis of microbiome composition and plant composition

In order to conduct a principal component analysis (PCA), we centre-log ratio (CLR) transformed numbers of reads for both diet and microbiome using the *tax_transform* function and carried out a PCA using the *ord_plot* function, both from the microViz package[106]. Samples are grouped by both

reserve and sampling season (pre- and post-rains) and the PCA shows loading vectors for the twenty plant taxa with the highest loading values.

Beta diversity is defined as the variability in species composition between sampling units, which here were individual dung samples. Using a PCA, it is possible to calculate the multivariate dispersion of a group of samples, demonstrate how dispersed they are along the principal component axes, and thus estimate how much beta diversity exists between samples in that group. We calculated beta dispersion among samples taken on the same reserve and during the same sampling season using the *betadisper* function within the microbiome package by calculating the mean Euclidean distance-to-centroid for all individuals sampled on a reserve and each sampling season. We performed an ANOVA and Tukey test to assess differences in beta dispersion.

Phylogenetic trees for each dataset were generated by first aligning sequences using the DECIPHER package[107]. We then constructed maximum likelihood trees using the phangorn package[108] for diet and microbiome phylogenetic trees. Phylogenetic trees were rooted at their midpoint using *mid.point* function in phytools[109].

We used the microeco package[110] to estimate weighted UniFrac beta diversity for both diet and microbiome. Weighted UniFrac indicates sample dissimilarity while controlling for phylogenetic distances between plant and bacterial taxa within samples[111]. We analysed compositional change using permutational analysis of variance (perMANOVA) using the adonis2 function in the *vegan* package[112] with 10,000 permutations. Firstly, we assessed whether there were differences in diet and microbiome beta diversity between populations and sampling seasons. We then tested whether diet composition changed across gradients of rainfall and NDVI, and for microbiome we tested how composition changed between populations in one model, and with NDVI and the relative abundance of the relevant plant families in another.

## Microbiome and diet shifts in Black Rhino

We calculated microbiome and dietary shifts using Euclidean distances between pre- and post-rain diet centroids for black rhino, at the individual and population level, and Grevy's zebra at the population level. Using the PCAs we calculated diet centroids as the average of the coordinates PC1 and PC2 for individuals with repeated sampling. We calculated the distance between centroids as:

$$\sqrt{\left( \left( PC1_{post-rain} + PC2_{post-rain} \right)^2 + \left( PC1_{pre-rain} + PC2_{pre-rain} \right)^2 \right)}$$

We tested whether individual black rhino seasonal dietary shifts predicted individual microbiome shifts using a linear model. We used the dietary shift for each breeding female in the breeding analysis. A linear model testing whether individual-level dietary shift correlates with mean relative abundances of plant families is presented in Supplementary Fig. 6 and Supplementary Table 5.

## Breeding and population performance

For Grevy's zebra, we used linear regression to test whether the mean proportion of grass in the diet was related to the percentage of foals in each Grevy's zebra population. For black rhino, we used a generalised linear model with a Gamma distribution to test whether seasonal dietary shifts affect females and affect inter-calving intervals.

## Reporting summary

Further information on research design is available in the Nature Portfolio Reporting Summary linked to this article.

## Data availability

The source data behind the graphs in the figures can be found in Supplementary Data 9. The raw metabarcoding output fasta files and some processed metabarcoding and sample data in xslx files can accessed on *Zenodo* zenodo.org/—Linking diet switching to reproductive performance across

populations of two Critically Endangered mammalian herbivores https://doi.org/10.5281/zenodo.10575034. Data regarding Kenyan black rhino and Grevy's zebra are treated as sensitive and confidential. There are therefore restrictions on the data that we can make available. Due to these confidentiality considerations, the sample data stored on *Zenodo* do not include locations of sample collection within each reserve for either species or the identity or breeding data for black rhino. The data also only include the final processed values for NDVI and rainfall. The remote sensing data are available from the repositories cited in the methods, but we cannot provide the shapefiles or other spatial data used to calculate the final values for each sample. Access to any of the data that are not accessible on *Zenodo* must be agreed upon by the individual study reserves. Readers must send data requests to the corresponding author and these will be passed on to reserve management teams. Please note that access to the data is likely to require research permits and a data use agreement.

## Code availability
All processing and analyses using QGIS version 3.16, Cutadapt 2.1 and R (v4.2.0) utilised standard code from the packages which are described in the methods and cited in full.

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

## Acknowledgements

We express sincere thanks to the security, monitoring and research teams on Lewa, Ol Jogi, Ol Pejeta and Mpala, and those at Grevy's Zebra Trust, who facilitated our fieldwork and sample collection. We would especially like to thank I. Lemaiyan, G. Chege, D. Kimiti, S.Nakito, and B. Gituku. We would also like to thank F. Omengo and KWS for providing permission to conduct this study and all the staff at Mpala Research Centre and ILRI who supported this work. The work of NHS and JB on this project was funded by the NERC Manchester-Liverpool DTP (NE/L002469/1), and they were also supported through the Chester Zoo Conservation Scholars programme. The work of NHS on the black rhino diet was also funded by the International Rhino Foundation (R-2019-9). SS is funded by a Royal Society URF (UF110641).

## Author contributions

N.H.S. contributed to conceptualisation, methodology, formal analysis, investigation, data curation, writing—original draft, writing—review & editing, visualisation, project administration and funding acquisition. J.A. contributed to methodology, formal analysis, investigation, data curation, writing—original draft, writing—review & editing and visualisation. R.A. contributed to methodology, formal analysis, resources and writing—review & editing. T.R.K. contributed to methodology, formal analysis, and writing—review & editing. D.I.R. contributed to formal analysis, data curation, and writing—review & editing. P.T. contributed to resources, writing— review & editing and project administration. B.E.K. contributed to investigation, resources and writing—review & editing. R.N. contributed to investigation, resources and writing—review & editing. D.H. contributed to investigation, data curation, and writing – review & editing. J.G. contributed to methodology, resources and data curation. S.M. contributed to methodology, resources and data curation. S.S. contributed to conceptualisation, methodology, formal analysis, investigation, writing—original draft, writing—review & editing, visualisation, supervision, project administration and funding acquisition.

## Competing interests

The authors declare no competing interests.

## Ethics approval

This project was approved by the University of Manchester's Committee for the ethical review of category D research (Ref: 0030). The research was conducted in affiliation with the Kenya Wildlife Service, which approved the

collection of samples, and licensed by the Republic of Kenya's National Commission for Science & Innovation (Permit numbers: NACOSTI/P/17/87006/16178, NACOSTI/P/19/1947 and NACOSTI/P/19/310). Kenyan scientists and conservation managers have been involved in this research from design through to authorship. This work is important for the management of both study species in Kenya, and the research questions were developed in conversations with reserve managers and the Kenya Wildlife Service (KWS) about the information required to improve conservation efforts. Reserve managers and rangers were fundamental to designing and carrying out the fieldwork protocol, and they provided the black rhino demographic data. Scientists at ILRI facilitated and contributed to the lab work. All the resulting data is available to the study reserves and KWS, and a summary of the conservation importance of the work will also be made freely available after publication. Five of the authors are at Kenyan institutions, and the work of other local conservation managers and scientists is mentioned in the acknowledgements. We cite research by Kenyan authors on both study species and the wider Laikipia ecosystem.
