## [Peer Review File · Communications Biology]

Reviewers' comments:

Reviewer #1 (Remarks to the Author):

Reviewer Comments COMMSBIO-23-2422-T Sky et al. Diet switching is associated with poor reproductive performance across populations of two Critically Endangered megaherbivores

In this paper the authors explore how indicators of breeding performance of black rhinos and Grevy's zebra may be influenced by diet switching, and changes in gut microbiome composition. Personally, I like to broad scale of using data from between study sites, seasonal changes in diet, changes in gut microflora and the implications of these things on fitness. Overall, the paper is well-written, but I think the story (i.e., the links between all the different measures) could be made a bit clearer. Moreover, there are a number of grammatical errors, but these are easily fixed. I am not an expert on the techniques used (e.g. DNA metabarcoding) so I can only really comment on the ecology. Nevertheless, the analysis seems correct, and the conclusions drawn justified. However, there a few things not fully explain that are set up in the introduction. Nevertheless, I believe that these too can easily be addressed. I have a few specific comments that I hope may further improve the manuscript.

Specific comments:

Page 1, Title: Megaherbivore refers to herbivores that are >1000 kg (see Owen-Smith 2002). Where black rhinos fall into this category, zebra do not. As such, 'megaherbivores' needs to be removed from the title. Maybe replace it with 'mammalian herbivores'?

Page 1 lines 33-34: The statement that 'change in microbes may impose a metabolic efficiency cost for instigating mismatches between diet and microbe composition' does not reflect what it reported in the text. This was not found for black rhinos and any link to a mismatch for zebra is not discussed. See additional point below.

Page 3, line 82: Maybe provide a couple of examples of the fallback foods of the primates. This would be interesting as many are likely omnivores.

Page 3, lines 102-104: Where the 'cost of switching hypothesis' is set up for both herbivores, conclusions are only really drawn for black rhinos. (See Page 9, lines 288-290). Some acknowledgment of whether it applies or not to zebra is required as currently this is not addressed (see Page 9, lines 291-297). This links back to the statement made in the abstract.

Page 5, line 155: RAA is not defined here. I understand that the format has the methods after the discussion, but shouldn't acronyms be defined the first time they appear in the text? This may be a simple style issue.

Page 5, lines 159-161: It would be good mention here that the genera within these plant families used by the two herbivores varied. For example, the extent to which the genera of Fabaceae were used by the herbivores is very different (i.e. use of woody Acacia by black rhinos and the use of the herbaceous perennials and annuals of Indigofera by zebra). Currently the reader needs to search the supplementary material (Tables S12, S13) to find this out.

Pages 7-8, lines 227-231: A strong negative, but not significant relationship? Quite simply the relationship is non-significant, correct? As such, just say it is non-significant and move on. You conclude that it is the proportion of Fabaceae in the diet that drives reproductive performance of zebra, so stick with that. However, if you are intent on focussing on the $P=0.068$, then say it is non-significant but that may be partially due to a small sample size. However, that would require further study.

Page 9, line 277: RS is not defined anywhere in the manuscript or supplementary material.

Page 11, line 339: 'herbivore diets'

Page 11 lines 348-349: 'Reserves be guaranteed to contain...' ??? I don't follow this. Please

reword.

Supplementary Materials

Figs S2-S7: I know that the study focusses on differences between the study sites, However, would it be possible to explore within study sites whether the different diets you recorded have led to differences in individual fitness (e.g. inter-calving interval or calf survival)? You suggest the possibility for zebra (See Page 10, line 310). You may only be able to do this for the individuals in one or two sites, but it would provide further support and take the findings down to a finer spatial scale. Just a thought.

Reviewer #2 (Remarks to the Author):

Dear Authors

I have enjoyed reading your manuscript which I believe provides an important addition to the field of foraging ecology and adds important links between foraging ecology, metabarcoding, microbiome composition and fitness consequences. It also provides results that highlight the current limitation when employing traditional methods (i.e. browse clipping) when assessing for example black rhino diet and hence habitat suitability. This study looked at the vegetation productivity (through NDVI), diet (through metabarcoding) and microbiome composition of two African herbivores (Grevy's zebra and black rhino) across a rainfall gradient and across a wet and dry season. It linked this to reproductive performance indicators and also to the microbiome composition.

Overall the paper is well written, but there are parts that require some clarifications. Attention also needs to be given to the tenses across the results and discussion. I attach an annotated document with detailed comments.

Reviewer #3 (Remarks to the Author):

The study titled, "Diet switching is associated with poor reproductive performance across populations of two Critically Endangered megaherbivores" evaluated the degree of diet switching that occurs in rhinos and Grevy's zebra across an ecological gradient, during periods of resource abundance and scarcity. The authors also looked at concurrent changes in the microbiome and the impact of the diet switching on reproductive performance in the two species. The manuscript investigates a topic of high interest to the field of ecology and presents valuable findings. The statistical analyses were rigorous and appropriate, and the discussion section was excellent. However, the manuscript could benefit from a bit of restructuring and streamlining to make ideas flow better and more coherent. At times, the points authors were making would get lost in unnecessary detail or verbose sentences. The method section could be reorganized as well to make the analyses easier to understand. Authors can also expand on their microbiome findings in the results section (and there are some microbiome findings mentioned in the discussion section that were missing from the results section). Please see my detailed comments in the attached document.

Response to reviewers

We would like to thank all three reviewers for their constructive comments. We are pleased that there is enthusiasm for the study but take note of the comments about structuring and wordiness. In this revision, we have made a concerted effort to streamline and reduce verbosity.

We have included all reviewer comments (apart from a few correction of very minor typos from reviewers 2 and 3) in this document. Line numbers refer to the new version of the manuscript without tracked changes, or in the tracked changes document with 'No markup' chosen under Tracking on the Review tab.

Reviewer 1

	Comment	Response	Revised text line numbers	Revised text
1	Page 1, Title: Megaherbivore refers to herbivores that are >1000 kg (see Owen-Smith 2002). Where black rhinos fall into this category, zebra do not. As such, 'megaherbivores' needs to be removed from the title. Maybe replace it with 'mammalian herbivores'?	Megaherbivores changed to 'mammalian herbivores' We have also changed the title in line with our response to comment 7. We are not claiming to provide evidence for a link between dietary switching and reproductive performance in Grevy's zebra.	Title	Linking diet switching to reproductive performance across populations of two Critically Endangered mammalian herbivores
2	Page 1 lines 33-34: The statement that 'change in microbes may impose a metabolic efficiency cost for instigating mismatches between diet and microbe composition' does not reflect what it reported in the text. This was not found for black rhinos and any link to a mismatch for zebra is not discussed. See additional point below.	See response to comment 4		
3	Page 3, line 82: Maybe provide a couple of examples of the fallback foods of the primates.	Examples included	71-75	Fallback foods have most often been studied in primates, and examples show that the

	This would be interesting as many are likely omnivores.			prevalence of these foods is associated with individual fitness proxies and population performance. Whilst the density of gibbons correlates with the density of their fallback food of figs ²⁴ , prolonged use of the corms of grasses and sedges can lead to longer interbirth intervals and aborted fetuses in baboons ²⁵ .
4	Page 3, lines 102-104: Where the 'cost of switching hypothesis' is set up for both herbivores, conclusions are only really drawn for black rhinos. (See Page 9, lines 288-290). Some acknowledgment of whether it applies or not to zebra is required as currently this is not addressed (see Page 9, lines 291-297). This links back to the statement made in the abstract.	The wording in the abstract has been changed to reflect the fact that the environment in both species NDVI, and in black rhino dietary composition, affects microbiome but seasonal dietary and microbiome shifts do not correlate. We have two sentences to show that the cost of switching hypothesis does not apply to zebra in the discussion (Lines 356-360). In light of these comments, we have also included a linear model to test whether individual level dietary shifts predict microbiome shifts (this is only possible for black rhino as we do not have the data available for zebra)	33-35 295-298 657-659	Abstract: Whilst microbiome composition between individuals was affected by the environment in both species, and diet composition in black rhino, seasonal dietary shifts did not drive commensurate microbiome shifts. Zebra did not seem to conform to patterns predicted by the cost-of-switching hypothesis as average reserve level seasonal dietary shifts did not match microbiome shift (Table S21). The large shifts were observed in the arid and grass-poor environment of Westgate suggests that one cost of reliance on fallback foods may be unstable microbiomes. Methods: As we had individual-level data for black rhino, we calculated individual seasonal dietary and microbiome shifts. We tested whether individual seasonal dietary shifts predicted individual microbiome shifts using a linear model.

		(methods: Lines 766-767, results: Lines 263-264). There was no significant effect which further evidences that this hypothesis does not apply to black rhino. We have also corrected a mistake in table S21 regarding average microbiome shifts for each reserve. Because of the new individual-level analysis and this corrected mistake, we have reworded the discussion of the cost of switching hypothesis for black rhino. This still shows that our results do not support it, but in a slightly different way (OI Jogi has the largest dietary shift, but has a microbiome shift in between Lewa and OI Pejeta). We have changed a sentence in the discussion to say that the prediction that pronounced seasonal dietary shifts lead to	212-213 299-305 306-311	Results: In black rhino, individual seasonal dietary shift was not correlated with microbiome shift ($\beta = 0.20$, $se = 0.19$, $t = 1.02$, $df = 51$, $R^2 = 0.00093$, $p = NS$). Our study does not support the cost of switching hypothesis for black rhino microbiomes either. However, in contrast to Grev's zebra, reliance on fallback foods did not seem to lead to microbiome instability as black rhino on OI Jogi consumed showed small shifts away from Fabaceae but had an average microbiome shift in between that of OI Pejeta and Lewa (Table S21). On Lewa and OI Pejeta, microbiomes became much more dispersed before the rains and overlapped more with those of OI Jogi black rhino, driven by shifts towards bacterial families including Planococcaceae, Moraxellaceae, Flavobacteriaceae, Weekselaceae and Dysgonomonadaceae (Table S19). It may be that seasonal shifts take place too quickly to significantly affect microbiome for any given individuals, and longer term dietary and environmental differences between individuals drive microbiome variation at a
--	--	--	--	--

		significant changes in the microbiome are not supported by our results rather than partially borne out in the discussion and that seasonal dietary shifts may be too fast to significantly affect microbiome.		population level. Whilst diet composition has been found to be a primary regulator of the microbial niche available in mammalian guts ⁶⁸ , microbiome composition is also driven by other factors. Secondary plant metabolites associated with certain taxa may significantly affect particular bacterial taxa ⁶⁹ .
5	Page 5, line 155: RAA is not defined here. I understand that the format has the methods after the discussion, but shouldn't acronyms be defined the first time they appear in the text? This may be a simple style issue.	RRA written out in full as 'relative read abundances' here. We also explain here that have substituted RRA for 'relative abundances' throughout the rest of the manuscript in response to comment 26 from reviewer 3.	142-143	The three plant families with the highest mean relative read abundances (which we will refer to as relative abundances)
6	Page 5, lines 159-161: It would be good mention here that the genera within these plant families used by the two herbivores varied. For example, the extent to which the genera of Fabaceae were used by the herbivores is very different (i.e. use of woody Acacia by black rhinos and the use of the herbaceous perennials and annuals of Indigofera by zebra). Currently the reader needs to search the supplementary material (Tables S12, S13) to find this out.	These suggestions have been added.	146-150	The highest utilised genera of Fabaceae and Poaceae differed between the two species. Within Fabaceae, black rhino consumed a high proportion of acacia (Vachellia and Senegalia) whereas zebra primarily consumed Indigofera . Among Poaceae, Cenchrus was the only genus that formed a high proportion of black rhino diets, whereas Grevy's zebra ate a lot of both Cenchrus and Digitaria .
7	Pages 7-8, lines 227-231: A strong negative, but not significant relationship? Quite simply the relationship is non-significant, correct? As such, just say it is non-significant and move on. You conclude that it is the proportion of Fabaceae in the diet that drives reproductive performance of zebra, so stick with that. However, if you are intent on focussing on the P=0.068, then say it is non-	Changed to say 'non-significant' here. Reference to this association has also been removed from the discussion and abstract, and the title has also been changed to reflect this (see comment 1)	238-240	There was a non-significant relationship between the proportion of legumes in the diet pre and post rainfall and the percentage of foals in each Grevy's zebra reserves

	significant but that may be partially due to a small sample size. However, that would require further study.			
8	Page 9, line 277: RS is not defined anywhere in the manuscript or supplementary material.	The sentence containing this was left in during the editing process by mistake and so has been deleted.		
9	Page 11, line 339: 'herbivore diets'	This typo has been corrected.	358	This work has important implications for future research into herbivore diets,
10	Page 11 lines 348-349: 'Reserves be guaranteed to contain...' ??? I don't follow this. Please reword.	This sentence has been reworded.	366-369	Protected area placement has often been controlled by social and economic factors ^{80,81} , rather than ecological ones such as optimal habitat for a target species. Even in reserves created for one flagship species, it cannot be guaranteed that the diet that the species is consuming there is optimal or near-optimal
11	Supplementary Materials Figs S2-S7: I know that the study focusses on differences between the study sites, However, would it be possible to explore within study sites whether the different diets you recorded have led to differences in individual fitness (e.g. inter-calving interval or calf survival)? You suggest the possibility for zebra (See Page 10, line 310). You may only be able to do this for the individuals in one or two sites, but it would provide further support and take the findings down to a finer spatial scale. Just a thought.	Thank you for the suggestion. Unfortunately we are unable to do this for zebra because we do not have individual-level data for zebra fitness, we just have the county-level data used in the analysis we have included. We do have individual-level data for rhino, and present an analysis of the effect of dietary shifts on individual inter-calving interval in Figure 7. We have tested whether dietary composition (eg. Proportion of dietary		

		Fabaceae/Poaceae) affects individual fitness but there are no direct significant relationships between composition and fitness.		
--	--	---	--	--

Reviewer 2

	Comment (line numbers from original manuscript)	Response	Revised text line numbers	Revised text
1	Do you mean that a core area is not necessarily near the centre then? Line 42	Wording has been changed, and a sentence moved from later in the paragraph to here, to clarify.	43-44	Core areas, with high ecological resilience, are not necessarily near the centre of a species' range ⁵ .
2	or rather than describe optimal habitats in order to prioritise their conservation? Line 45	Clarification added to this sentence.	44-47	Identifying and prioritising resilient populations, and ecological characteristics that describe optimal habitats, are key for predicting and arresting biodiversity loss ^{6,7} , especially where a species' range is dynamically changing in response to environmental change ⁸⁻¹⁰ .
3	did you find a correlation between NDVI and rainfall i.e.how were you able to have both in the models Line 150	Both of these variables are used as predictors in in the diet composition permANOVAs, and the linear models of the diet principal components (ie do PC1 and PC1 depend on NDVI and rainfall).		

The graphs here show the relationships between rainfall and NDVI for the rhino and zebra samples separately. Whilst NDVI is dependent on rainfall, the relationship between the two is different between reserves. This can clearly be seen for zebra and for the rhino in a polynomial linear model $NDVI \sim Rainfall^2 * Reserve$ there is a significant interaction between Rainfall and Reserve.

Whilst NDVI does depend on rainfall, we believe that rainfall is affecting diet independently of its

		effect on greening, by for example driving changes in the nutrient content of evergreen plants and/or altering movement patterns (animals may be able to move further from water points and access a more varied diet if ephemeral water sources become available or they can get more water from their food). If there were not independent effects of rainfall and NDVI then they should not both be significant in the relevant models, in which case we would not have retained them both. We therefore believe that each of the two variables are having independent effects on diet, so believe that it is valid to keep both in the models.		
4	More than what? Line 237	Clarification added to this sentence.	247-250	Optimal foraging models predict that diet diversity should increase with resource scarcity as valuable foods become less abundant ²¹. However, we found more nuanced relationships between dietary responses, foraging strategy and vegetation productivity gradients than predicted by OFT.
5	lower than what? AND I am not following what you mean by this sentence and the reference to knowing what proportion migrates Lines 245+246	This sentence has been deleted, and a clearer description of these ideas has been added. The part of the discussion regarding implications for the management of both species has been reordered to improve flow (Lines 320-332)	325-338	When environments become dry and NDVI decreases, Grevy's zebra can migrate to find more suitable conditions ^{70,71} resulting in seasonal dispersal dynamics across the metapopulation. This could be within a reserve, as we found large differences in the proportion of grass consumed by individuals on the same reserve, particularly during the dry seasons on

				Lewa and Mpala-Ol Jogi (Figure S8) indicating variable access to grass across these reserves, or across larger distances. Historically, zebra were able track grass availability, which allows individuals to buffer their diets. The Laikipia-Samburu landscape is under pressure of fragmentation from fencing ⁷⁶ with conservancies in the south and the west being disconnected from those in the central and northeastern regions ⁷⁰. Limiting the ability of grazing species, including Grevy's zebra, to seasonally track resources is likely to result in greater reliance on fallback foods, more frequent and pronounced diet switching and decreased resilience of their metapopulations. Future research, particularly on Westgate, to see what proportion of the population migrates, what proportion stays in place, how this varies by age and sex, and how fitness varies between the strategies would be important for conservation.
6	when green or when? Line 265	Clarification added. It must be noted that the study referenced here is not clear about exactly when in the seasonal cycle these high proportions of C₄/CAM plants are found in rhino diets.	278-280	This is supported by stable isotope ratio research that showed some black rhinos in Laikipia have up to 40% of their diets made up of C₄/CAM plants for short periods, which is thought to occur after rains when grasses have the highest level of nutrition⁵⁹.

7	Comments to change present tense to past tense throughout discussion	Tense changed throughout discussion so it is all in past tense		
8	which models are you referring to here? Line 270	Clarification added	285-287	Theoretical models from other research suggest that the ability to change diets seasonally is advantageous for savanna herbivores and that mixed feeders have access to more and better food than specialised browsers or grazers ²⁰ .
9	unfinished sentence Line 276	This sentence has been deleted (this is the same change as in response to reviewer 1 comment 8)		
10	compared to? Line 326	Clarification added.	343-346	Previous work has shown that female rhino on OI Jogi exhibit lower breeding rates and higher mortality rates than Lewa and OI Pejeta, and the population is more susceptible to extinction than in the other two reserves ⁷⁷ where Fabaceae is a smaller component of the diet.
11	I am not following this sentence and how it flows from the previous one Line 349	This has been reworded to make this clearer (this is the same change as in response to reviewer 1 comment 10)	366-369	Protected area placement has often been controlled by social and economic factors ^{80,81} , rather than ecological ones such as optimal habitat for a target species. Even in reserves created for one flagship species, it cannot be guaranteed that the diet that the species is consuming there is optimal or near-optimal ⁸² .
12	could you say something about wet season rainfall vs dry season and timing of the two Lines 376-377	Sentences added at the end of this paragraph	406-413	This region of Kenya traditionally has two annual rainy seasons; the long rains March-May and short rains October-December. Monthly rainfall peaks at around 100 mm in April and November on Lewa and OI Pejeta, and

				around 80mm on Mpala and Ol Jogi. Mpala, Ol Jogi and Ol Pejeta have rainfall relatively well-spread throughout the rest of the year, with around 20-50 mm per month and a small peak in August, while Lewa has a pronounced dry period July-September⁵⁶. Westgate has a similar timing of rains but lower amounts at all times of year compared to the other reserves. It should be noted that Kenya has been experiencing a drought since 2020, but our samples were collected before this disruption to rainfall patterns.
13	what type of samples? This is not clear enough AND How do you estimate a home range with n=4 maybe clarify that you were not interested in the exact home ranges but in a way to delimitate the possible area used by the rhinos to get NDVI info Line 404`	This has been reworded to make it clearer. We have also added the suggested clarification regarding these areas not being exact home ranges, but rather an estimate of the possible area used by the rhinos. To reflect this, we have changed 'home range' to 'utilised area' throughout the manuscript.	428-433	We extracted NDVI in areas we estimated were utilised by each individual. Black rhino home ranges vary in size and are generally larger than the resolution of the NDVI data ⁵⁷. We were not interested in exact home ranges but rather a way to estimate the possible area used by individual black rhinos across the sampling periods. We estimated this utilised area using the dung samples used in this study, and more which were collected using the same method but were not included in the metabarcoding analysis.
14	And what about zebra? AND this is unclear. SO you used to different methods for the two species?	This has been clarified. We did use different methods for the two species. As all analyses were done separately for each species, this does not pose a problem as	443-446	We did not have definite repeat samples for Grevy's zebra, so used a different method to estimate the area of utilisation. As gut retention time for

	Line 412 and 415	these estimation were done consistently within each species.		We did not have definite repeat samples for Grevy's zebra, so used a different method to estimate the utilised area. As gut retention time for Grevy's zebra is approximately 24-48 hours and Grevy's zebra can move ~2.3 km per day ⁶¹ , the mean NDVI was calculated within a circle with a radius of 2.3 km centred on the sample collection point.
15	How collected? Already available? From where?	Information to answer these questions added to the start of the paragraph.	453-457	The three black rhino reserves record the dates of births, deaths, imports (including the ages of imported individuals) and exports as part of their regular monitoring. The ages at which females died and gave birth has been accurately recorded since the foundation of each sanctuary. The reserves supplied these records and we formatted them so we could calculate what we called female inter-calving interval.
16	over the age of five years, calculated in months line 419	This suggestion has bee included	457-458	This is the age of the female over the age of five years, calculated in months, divided by the number of calves produced.
17	What measure? AND I am not following this Lines 421 and 423	Changes made to improve clarity	460-463	Other studies have estimated breeding success using the age of first calving, and a measure of inter-calving interval calculated by dividing the number of months between a female's first calf and the present/death by the number

				of calves produced. Our measure of female inter-calving interval incorporates information from both of these measures.
18	So only females were sampled? Line 440	No males and females were sampled, we don't say that only females were sampled but this sentence has been reworded so hopefully this is clearer. We have also removed the reference to taking photographs of zebra as it is not relevant to this study.	480-482	For zebra, samples were collected from individuals that had been observed defecating. Several whole droppings were taken from each dung pile. We photographed and recorded the sex of sampled individuals.
19	Wiener Line 595	Correction to the name of this index has been made	609-610	We calculated Shannon-Wiener index of dietary alpha diversity using the estimate_richness function in the phyloseq package
20	Not for zebra? (Referring to the presentation of AIC values) Line 598	We have now included AIC values for all relevant zebra models as well, and state this in the methods. AIC values are presented in relevant tables	606-608	For these and all other linear and linear mixed effects models we present AIC values which were calculated as part of the glm function for Grevy's zebra and using the AIC function for black rhino.

Reviewer 3

It is worth saying that we very much appreciated the thoughtful suggestions that reviewer 3 made to help improve clarity and flow of the writing.

	Comment (line numbers from original manuscript)	Response	Revised text line numbers	Revised text

1	The first two sentences of the abstract (L22-26) do not accurately communicate the complexity of the study, the study's questions, or the gap in the literature that is being addressed. I suggest the authors write something along the lines of: "Optimal foraging theory predicts that animals will maximize their energy intake by consuming the most valuable foods available. Animals will expand their diets to include lower quality "fallback foods" when resources are more limited, which may impact fitness. However, it is relatively unknown the degree of diet switching and incorporation of fallback foods that occurs in herbivores across ecological gradients and seasonal conditions. Here we evaluated temporal diet switching across populations of Critically Endangered eastern black rhino (Diceros bicornis michaeli) and Grevy's zebra (Equus grevyi) before and after rainfall periods in Kenya. We also determine whether dietary shifts are associated with habitat quality, variation in the microbiome, and reproductive performance in the two species." It is important that the authors mention foraging theory, fallback foods, and what is unknown (and motivated the current study).	The abstract has been rewritten, and we have used the suggested sentences with a few modifications. Please note that the abstract is currently 200 words. We are aware the journal recommends 150 but we will not be able to include all the information requested by the reviewers within that limit. If this is a problem please let us know and we can redraft.	22-37	Optimal foraging theory predicts that animals maximize energy intake by consuming the most valuable foods available. When resources are limited, they may include lower quality 'fallback foods' in their diets. As seasonal herbivore diet switching is understudied, we evaluated its extent and effects across three Kenyan reserves each for Critically Endangered eastern black rhino (Diceros bicornis michaeli) and Grevy's zebra (Equus grevyi), and its associations with habitat quality, microbiome variation, and reproductive performance. Black rhino diet breadth increased with vegetation productivity (NDVI), whereas zebra diet breadth peaked at intermediate NDVI. Black rhino diets associated with higher vegetation productivity had less acacia (Fabaceae: Vachellia and Senegalia spp.) and more grass suggesting that acacia are fallback foods, upending conventional assumptions. Larger dietary shifts were associated with longer calving intervals. Grevy's zebra diets in high rainfall areas were consistently grass-dominated, whereas in arid areas they primarily consumed legumes during low vegetation productivity periods. Whilst microbiome composition between individuals was affected by the environment, and diet
---	---	--	-------	--

				composition in black rhino, seasonal dietary shifts did not drive commensurate microbiome shifts. Documenting diet shifts across ecological gradients can increase the effectiveness of conservation by informing habitat suitability models and improving understanding of responses to resource limitation.
2	L27: replace “NDVI” with “vegetation productivity (NDVI)”	Done	27-28	Black rhino diet breadth increased with vegetation productivity (NDVI), whereas zebra diet breadth peaked at intermediate NDVI.
3	The introduction is long and contains details that are at times not required. As a result, the important messages that the authors are communicating may get lost in the text. Trimming out the unnecessary detail will help streamline ideas and make it more obvious to the reader what purpose of study is and what questions are being addressed. AND I suggest authors comb through the introduction and make similar types of edits and deletions.	We have done this, all changes can be seen in the revised manuscript version with tracked changes.		
4	Take the following sentences (L51-59) (my additions are in orange) The crossed-out sentences are not critical to understand the study and by deleting them, authors can make sure readers	We have made the suggested changes.	51-59	Fitness is directly associated with maintaining sufficient energy reserves to support metabolism, invest in reproduction, and buffer periods of scarcity ¹² . During periods of scarcity,

	understand the important takeaway of this paragraph which was the significance of fallback foods and their impact on herbivores, and how diet switching has been poorly investigated.			animals can move to follow changing distributions of valuable foods, alter their foraging strategies, shift their diets, or any combination of those. Dietary changes can cause energetic stress particularly when animals switch to consume less preferred ‘fallback foods’ during times of scarcity¹³ that they may not be physiologically adapted to digest^{14–17}. Diet-switching often occurs between high and low vegetation productivity periods (i.e., summer/winter or before/after rains) in grazers such as bison (Bison bison)¹⁸ as well as mixed-feeders such as moose (Alces alces)¹⁹. Despite its potential importance, the role of dietary strategy in driving herbivore population dynamics is poorly understood²⁰.
5	I have a similar comment regarding the following sentences (L65-78). Removal of the crossed-out sentences greatly improves the cohesion of the paragraph and makes it much easier to follow the author’s logic and statements. The transitions are more seamless between fallback foods, differences between guilds, and the impact of diet switching on reproductive success.	We have reworded using them as a guide. We have moved the information on herbivore guilds to later in the introduction to improve flow. We have retained some information that was suggested to be deleted from the second paragraph, mainly the sentence regarding intraspecific comparisons, because we believe it is vital to understand why we have used the chosen method. But we have incorporated some information from the end of the paragraph, and deleted a sentence there, which streamlines it overall.	61-71	Optimal Foraging Theory (OFT) can predict the composition of a species’ diet under particular conditions. During periods of abundance, herbivore diets should contain relatively few species as animals maximize their energy intake by concentrating on the ‘best’ food plants available in terms of energy and nutrients. When and where valuable foods are more limited, animals should expand their diets to include lower quality fallback foods^{19,21}. Whilst OFT can help predict what foraging strategy

				is optimal under a given set conditions, it cannot indicate the impact that each diet will have on survival or reproductive success. Intraspecific comparisons of the frequency with which diet-switching is required across ranges ²² and corresponding demographic indicators can provide a mechanistic explanation for how ecological conditions lead to demographic heterogeneity. Fallback foods can be identified by heavy use during periods of scarcity, or by research into nutritional qualities ^{23,24} .
6	L66: change “plenty” to “abundance”	Done		
7	L81-82: This sentence has awkward, and confusing phrasing, please revise. The confusion mostly lies in the usage of “an be identified by a negative correlation” and “and/or independent knowledge of”	Wording has been changed to improve clarity	69-71	Fallback foods can be identified by heavy use during periods of scarcity, or by research into nutritional qualities ^{23,24} .
8	L97: Can delete the first part of that sentence and start directly with “Diet switching may also impose a cost on metabolic efficiency in herbivores.”	Done	88	Diet switching may impose a cost on metabolic efficiency.
9	L116: delete with “with green up associated”	Done	100-102	East African savanna ecosystems present an excellent opportunity to evaluate seasonal diet switching because rainfall is seasonally concentrated such that the diversity and biomass of both herbaceous and deciduous plants increases with the onset of rains ⁴⁶⁻⁴⁹ .
10	L123: add (e.g., trees and shrubs from Ebanaceae) after “all other families”	Done	109-111	Savanna herbivore diets can be described along three axes using the proportions of legumes (Fabaceae

				family), grasses (Poaceae family) and all other families (e.g., trees and shrubs from Ebenaceae) ⁵⁴ .
11	L127-131: I suggest the authors delete these three sentences as they are not critical for understanding the study since this study did not focus on or measure competition. Sometimes these extra details interrupt the flow of paragraph and dilute the important message of this paragraph which is that we do not know the extent to which zebra and rhino diet switch and use fallback foods, and how this impacts fitness. AND L131-134: Can delete this sentence, as it repeats information mentioned earlier in this paragraph. Authors already explained the plant preferences of the two herbivore species	Done. The rest of this paragraph has also been moved down and the following paragraph split into two.	107-117	We evaluated temporal diet switching and microbiome variation across five reserves containing two IUCN Red List Critically Endangered savanna herbivores with different foraging strategies; the eastern black rhino (Diceros bicornis michaeli) and Grevy's zebra (Equus grevyi). Savanna herbivore diets can be described along three axes using the proportions of legumes (Fabaceae family), grasses (Poaceae family) and all other families (e.g., trees and shrubs from Ebenaceae)⁵⁴. Whilst black rhino diets do vary seasonally^{58,59} they are considered to be browsers, to find woody plants in the family Fabaceae particularly valuable⁵⁵⁻⁵⁷, and to ideally have herbaceous plants, including grasses, contribute little to their diets⁵⁸. Grevy's zebra, in contrast, are grazers with a preference for grasses and other herbaceous vegetation⁶⁰⁻⁶². We do not know, firstly the degree to which animals incorporate fallback foods seasonally and secondly whether dietary switching has implications for fitness. We seek to provide evidence relevant to both questions.

12	L147: add “variation in the microbiome” after “variation in habitat quality” AND L147: edit “by these large herbivores” to “in these large herbivores”	Done	124-125	Second, to determine whether dietary shifts and the use of fallback foods is associated with variation in habitat quality, variation in the microbiome or calving rates in these large herbivores.
13	The methods were a bit disjointed and had an overuse of subheadings that would break up the flow and make it harder to follow what was being written. These headings would sometimes provide a separation between content that did not need to be separated. Sometimes, subheadings were provided but were not really needed (e.g. the subheadings “Amplification of bacterial DNA”, “Amplification of plant DNA”, “Pool normalization”).	We have reduced the number of subheadings used in the methods, the subheadings referenced here have been deleted and all metabarcoding methods fall under the heading ‘DNA extractions for the metabarcoding of the 16S rRNA gene and chloroplast trnL genes’ (See comment 18)		
14	I recommend the authors restructure the Analysis section to make it easier to navigate. Instead of having 6 subheadings, maybe it can contain only four: 1. Diet analyses (can include the dietary breadth Shannon analyses & the RRA correlations since these are all related) 2. Beta-diversity analysis of microbiome composition and plant composition 3. Microbiome and Diet shifts in Black Rhino 4. Breeding and population performance	Done		
15	I also advise the authors provide a few sentences at the beginning of this Analysis section summarizing the analyses that were conducted and that are about to be explained in the following subsections.	Done	584-589	Analyses were used to test what environmental factors affected diet and microbiome composition, what plants were favoured during times of abundance, whether important plant

				families were substituted for each other seasonally, and whether seasonal diet shifts drove changes in microbiome. We also tested whether the proportion of grass in the zebra diets, and the size of seasonal dietary shifts in black rhino, affected demographic indicators of fitness.
16	L369: Edit “Samples were collected at five sites:” to “Faecal samples from rhinos and Grevy’s zebra were collected at five ecological reserves in Kenya.” Include the years of sample collection.	Done, except for the substitution of ‘reserves’ for ‘ecological reserves’ (see comment 28)	386-389	Faecal samples from rhinos and Grevy’s zebra were collected at five reserves in Kenya in 2018 and 2019: Ol Pejeta Conservancy (black rhino), Ol Jogi Conservancy (black rhino and Grevy’s zebra), Lewa Wildlife Conservancy (black rhino and Grevy’s zebra), Mpala Research Centre (Grevy’s zebra) and Westgate Community Conservancy (Grevy’s zebra) (Figure S1).
17	L86: Can authors provide a bit more description on Commiphora , Balanites , Boscia and Grewia ? Are they woody plants, grasses, legumes?	Added.	402-406	Westgate is in Samburu County (0.81°N, 37.3°E), has an average annual rainfall of around 190 mm and is savanna grassland with varying densities of Vachellia , Commiphora (woody shrubs and trees in the Burseraceae family), Boscia (woody shrubs and trees in the Capparaceae family) and Grewia (woody shrubs and trees in the Malvaceae family) ⁸⁹ .
18	L448: “Metabarcoding” header can be renamed to “DNA extractions for the metabarcoding of the 16S rRNA gene and	This section includes the protocol for more than just extraction so we have modified the suggestion slightly	494	DNA metabarcoding of the 16S rRNA gene and chloroplast trnL genes

	chloroplast trnL genes.” The authors can then delete the subheadings “extraction”, “amplification of bacterial DNA”, and “amplification of plant DNA.”			
19	L446: Can authors include a brief sentence about how fecal storage in ethanol impacts microbiome profiles?	Added with relevant reference - Blekhman, R. et al. Common methods for fecal sample storage in field studies yield consistent signatures of individual identity in microbiome sequencing data. Sci. Rep. 6, 31519 (2016).	489-492	Comparison of sequencing data obtained from faecal samples immediately frozen to those stored in ethanol has shown that while diversity metrics can be altered by storage methods, biologically relevant individual microbiome identity was retained ⁹⁹ .
20	L464 and L471: Please include the primer sequences	Included in SI (Tables S1-S3)	514	Primer sequences can be found in the Supplementary Information (Tables S1-S3).
21	L504-512: Can authors explain a bit more on the purpose of the full sequencing? I am not understanding why it was performed.	The full sequencing is the part of the protocol that produces the analysable data. This confusion probably arose because we did not explain the titration run in enough detail. We have done that and added a sentence clarifications to explain the purpose of the full sequencing run as well.	535-537 544-549	Sequencing bias is reduced by having roughly equal concentrations of DNA from each sample in the sequencing pool for the full sequencing run, which produces the analysable data. We used a titration sequencing run to estimate the amount of tagged DNA sequences present in each sample. The results of the titration sequencing were used to create the pool for the full sequencing runs which produce the analysable data. the amount of each product included in the pool for the final sequencing run was inversely proportional to the occurrence of their tagged sequences in the titration run.

				This equalises the amount of DNA from each sample as far as possible. For the full sequencing runs, we determined the average fragment size using TapeStation, quantified the concentration on the Qubit, and used these to dilute the pool to 4nM.
22	L537: Can the authors move the phylogenetic tree paragraph to the analysis section when talking about Unifrac? I am assuming this is what the trees were used for? Were they used for other analyses as well?	Done. The reviewer is correct, that it is the only analysis they were used for.		
23	L605: How did the authors control for multiple comparisons when conducting the Spearman correlations?	Thank you for this suggestion. We have now applied Holm corrections for these correlation tests as described in the revised methods. We have used Holm because our hypothesis tests are not fully independent and we also don't necessarily expect the different p values to be positively correlated, which Hochberg and Hommel rely on. We could have used false discovery rate correction, but we have chosen Holm as it is more stringent. For Grevy's zebra both results remain significant. For black rhino results of comparisons with all individuals remain the same, but two of the reserve-level comparisons have changed from significant to non-significant (Fabaceae vs Poaceae on Lewa, and Ebenaceae vs Poaceae on Ol Pejeta). We have made corresponding alterations to the results, table S18 and discussion.	618-621 179-193	Methods: Correlations for all individuals are presented in the main text, and for each individual black rhino reserve are presented in the supplementary information. Holm corrections were used separately for the black rhino and zebra tests to control for multiple comparisons using the p.adjust function. Results: For both species, there appeared to be a trade-off between grass and legumes (Figure 6). In black rhino, the proportion of Poaceae in the diet decreased with increasing Fabaceae (Spearman's rank correlation with Holm correction: $r = -0.30$, 95% CI = lower = -0.42, upper = -0.17, $t = -4.59$, $p < 0.001$, $df = 214$, Figure 6) and the proportion of Fabaceae decreased with increasing

			Ebenaceae ($r = -0.56$, 95% CI = lower = -0.64, upper = -0.46, $t = -9.76$, $p < 0.001$, $df = 214$, Figure 6), but there was no correlation between Poaceae and Ebenaceae. At a reserve level, the correlation between Fabaceae and Poaceae was only significant on Ol Pejeta. (Table S18). Grevy's zebra switched to a legume (Fabaceae) based diet when they could not saturate their diet with grass. Individuals in Lewa and Mpala-Ol jogi maintained the highest level of grass consumption across seasons. Individuals in Westgate did not display a period during our study where the average diet was majority grass (>50%). In Grevy's zebra across reserves the proportion of Poaceae increased with decreasing Fabaceae (Spearman's rank correlation with Holm correction: $r = -0.93$, 95% CI = lower = -0.97, upper = -0.87. $p < 0.001$, $df = 153$, Figure 6). Indigofera spp were the major alternative food type consumed by Grevy's zebra across reserves and seasons ($r = -0.91$, 95% CI = lower = -0.95, upper -0.85. $p < 0.001$, $df = 153$). 261-265 Discussion: It is likely that individuals employ different strategies, with some able to access valuable foods year-round, some switching seasonally, and others constrained to fallback foods by
--	--	--	--

				their ranges. This would explain why Fabaceae and Poaceae are negatively correlated for all individuals but not on Lewa and Ol Jogi, as more individuals on these reserves seemed to be constrained by lack of access to Poaceae in even after the rains (Figure 6).
24	Please include a few sentences to remind readers of the purpose of study, and the questions that were asked. It would be helpful to mention the main types of analyses that were done (e.g. dietary analyses, microbiome analyses, fitness analyses).	Done	127-131	In order to connect seasonal dietary shifts across an environmental gradient to changes in the microbiome and fitness we carried out four main analyses. For each sampled individual we characterised the change in seasonal vegetation productivity (NDVI) in their surrounding area, assessed the composition of diet and microbiome, assessed the drivers and extent of seasonal shifts, and finally connected seasonal dietary shifts to indicators of female breeding success.
25	L151: add “vegetation productivity” before (NDVI). Readers not familiar with this acronym would have forgotten what it stands for.	Done		See text for comment 24
26	L155: authors can say “relative abundances” instead of “RRAs” throughout the manuscript; this is more intuitive for readers.	Done		
27	L156: Can the authors remind readers of what each plant family encompasses? For example, “The three plant families with	Done	142-145	The three plant families with the highest mean relative read abundances (which we will refer to as relative abundances)

	the highest mean relative abundances across reserves and seasons for black rhino were Fabaceae (woody plants & legumes; mean of 45%), Ebenaceae (evergreen trees and shrubs; mean of 23%) and Poaceae (grasses; mean of 10%).”			across reserves and seasons for black rhino were Fabaceae (woody plants and legumes; mean of 45%), Ebenaceae (evergreen trees and shrubs; mean of 23%) and Poaceae (grasses; mean of 10%).
28	L159: Would it make more sense to use “ecological reserves” instead of “populations” since that is essentially what authors are referring too? That way terminology is consistent throughout the analyses? Currently, this predictor is referred to as “reserves” for some analyses and “populations” for other analyses.	We have changed terminology to use the word ‘reserve’ throughout instead of ‘population’. We have not seen the term ‘ecological reserve’ used that commonly in eastern Africa, and the terms ‘nature reserve’, ‘wildlife reserve’, ‘wildlife conservancy’ etc seem to be used to differing extents around the world. As a compromise, we feel that ‘reserve’ will be generally understood by all in the context of this work.		
29	L162: This paragraph references Fig 2a and 2c but Figure 2 only has one panel (a boxplot of NDVI)	The boxplot has been altered to split up black rhino and zebra results. Doing this also made us realise that we had not described the methodology of the analysis that shows differences in NDVI between reserves and seasons adequately, and we could do it in a robust way. As this is done at the ‘home range’ (what we now call area of utilisation in response to reviewers’ comments) level for each species we have split this up into two analyses, one for rhino and one for zebra. NDVI is now analysed using a linear regression for zebra, and a linear mixed effects model with ID as a random effect, for rhino. This now matches other analyses in the manuscript, including those for alpha diversity.	447-451	Methods: We analysed whether NDVI varied between reserve and sampling season at the area of utilisation level separately for each species. We used linear regression for the zebra, and linear mixed effects models, with black rhino ID as a random effect, for the black rhino. For both models we log-transformed NDVI as the response variable because NDVI was right skewed.

			133-140 Results: For black rhino (linear mixed effects model: standard deviation of random effect = 0.015, AIC = -1098.91) NDVI increased from pre-rain to post-rain ($\beta = 0.073$, se = 0.0021, $t = 34.62$, df = 150.8, $p < 0.001$) was lowest on Ol Jogi and highest on Ol Pejeta (Lewa to Ol Jogi: $\beta = -0.022$, se = 0.0033, $t = -6.67$, df = 60.7, $p < 0.001$, Lewa to Ol Pejeta: $\beta = 0.030$, se = 0.0038, $t = 8.00$, df = 69.3, $p < 0.001$). For Grevy's zebra (linear regression: $R^2 = 0.80$, df=150), NDVI also increased pre-rain to post-rain ($\beta = 0.029$, se = 0.0028, $t = 10.02$, $p < 0.001$), although there was not much seasonal change on Westgate. NDVI was lowest by far on Westgate and highest on Lewa (Lewa to Mpala-Ol Jogi: $\beta = -0.013$, se = 0.0034, $t = -3.80$, $p < 0.001$, Lewa to --	--	--	---

				Westgate: $\beta = -0.078$, $se = 0.0038$, $t = -20.66$, $p < 0.001$).
30	L172: Add “grasses” after “Poaceae” and “legumes and woody plants” after Fabaceae. Add “(evergreens)” after “Ebanaceae.” This may seem redundant, but I believe it is necessary because a large focus of this paper is on these 3 bacterial families and non-plant experts may forget what these families are.	Done	166-171	Across reserves within both species, the proportion of Poaceae (grasses) in the diet increased with increasing NDVI but saturated at different proportions (Table 2, Figure 4). Across all reserves, the proportion of Fabaceae (legumes and woody plants) in the diet of black rhino and Grevy’s zebra decreased with increasing vegetation productivity (Table 2, Figure 4). The proportion of Ebenaceae (evergreen trees and shrubs) in the black rhino diets was not associated with vegetation productivity (Table 2).
31	L202-205: It would be valuable if the authors expand this paragraph and list the most abundant bacterial genera in the microbiomes of rhinos and zebras. Were there any differences in the composition between the two species? How did composition change with season (did any genera or families increase or decrease)? The authors can consider including plots of microbiome composition in the supplementary materials. Furthermore, in the discussion section (L291-297), the authors mention that several bacterial families were associated with vegetation productivity. However, this result is not	We have added some sentences describing the most abundant bacterial taxa, and both seasonal and species differences in the results (lines 246-255) The key findings of this paper relate to intra-species differences across geographic space. Microbiome differences across large mammalian herbivores have been described previously (Kartzinel et al, 2019). We have therefore opted to not compare microbiome composition between species to ensure that our key findings are highlighted. Our figures within the main text highlight seasonal differences between reserves in both species. However, the reviewers makes a good point that we have not outlined all the data	195-204	Grevy’s zebra microbiomes are dominated by Lachnospiraceae, which have a slightly higher relative abundance on Lewa and Mpala-Ol Jogi of 20-25% than Westgate. No bacterial taxa is as prevalent in black rhino microbiome, with the order WCHB-41 having the highest average relative abundance of around 15%, although this is highest on Ol Pejeta post-rains with an average of 21%. This order is the second most common on average in zebra. The most common bacterial taxa are similar between both species, although Christensenellaceae and

	found in the results section, and they sounded very interesting! These should be added to the results section.	within these figures sufficiently. To accommodate this we have added key species' loading scores to the results section to highlight species which load onto important PC axis relatively strongly and have expanded the mention of bacterial families associated with vegetation productivity. (271-276 and 282-285). We have also added loading score tables into the supplementary material to highlight the bacterial families which are associated (and hence increase or decrease) with PC axis (tables S22-S24). These data were derived from loading scores hence if a bacterial family loaded significantly onto an axis which correlated with NDVI then we suggest that they are associated. However, identifying exact dynamics or co-occurrence of specific bacterial families in detail is outside the scope of this manuscript. It would however be an interesting future project. References: Kartzinel, T.R. et al. (2019) 'Covariation of diet and gut microbiome in African megafauna', Proceedings of the National Academy of Sciences, 116(47), pp. 23588–23593. doi:10.1073/pnas.1905666116.	219-225 231-234	Oscillospiraceae are more common in Grevy's zebra, whereas Spirochaetaceae and Moraxellaceae are more common in black rhino. There are no clear seasonal changes in the relative abundance of the most common bacterial taxa across reserves, except for Spirochaetaceae and Moraxellaceae in black rhino which both decrease post-rains across all three reserves. For black rhino, bacterial families Ruminococcaceae, Anaerovoracaceae, Anaerovoracaceae, Desulfovibrionaceae, Acidaminococcaceae (loading scores > 0.7) negatively loaded onto PC1 while Xanthomonadaceae, Flavobacteriaceae, Comamonadaceae, Weeksellaceae, Planococcaceae positively loaded onto PC1 (Table S22). Furthermore, rh-aaj90h05, Izemoplasmatales, Methanocorpusculaceae, Dysgonomonadaceae, f082, planococcaceae, mycoplasmataceae, ucg-010 and anaerovoracaceae positively loaded onto PC2 (loading scores > 1.0) (Table S23). Bacterial families including Planococcaceae, Paludibacteraceae, Bacteroidales bs11 gut group, Ruminiclostridium, Enterobacteriaceae, Desulfovibrionaceae Bacteroidales rf16
--	---	--	-------------------------------	--

				gut group and Campylobacteraceae negatively loaded onto PC1 (Loading scores >1) and are therefore associated with high NDVI and grass in the diet (Table S24).
32	Fig 7: Edit axes font sizes so that they are the same size in panel a and b	Done		
33	I could not find references to Table S11,S15,S16, and S17-19 in the main manuscript text. Can the authors reference these Tables in the main text so that readers know these exist and can be found in the supplementary material? Same comment for references to Fig S2 and Fig S8 (they are missing in the main manuscript text).	We have added in extra items into the SI and reordered them, so the numbers of the tables and figures that this comment refers to have changed, but we have ensured that all SI items are now referred to at least once in the main manuscript text.		

Other corrections

Correction of description of relationship between Poaceae RRA in zebra diet and dietary breadth as a negative quadratic, not negative binomial (line 182).

Change to reporting of zebra breeding model to make it consistent with the reporting of other linear regressions in the text. The actual statistics have not changed (lines 246-247).

REVIEWERS' COMMENTS:

Reviewer #1 (Remarks to the Author):

Reviewer Comments COMMSBIO-23-2422A Sky et al. Linking diet switching to reproductive performance across populations of two Critically Endangered mammalian herbivores

The authors have done a very good job of addressing my comments and suggestions. I look forward to see this published.

All the best
Adrian M Shrader
Mammal Research Institute
Department of Zoology & Entomology
University of Pretoria
South Africa

Reviewer #2 (Remarks to the Author):

Dear Authors

Firstly I would like to commend you for the very clear and detailed response to reviewers documents. It clearly shows how carefully you considered all the feedback you have received. This is now a very well written and executed study, well done!

Reviewer #3 (Remarks to the Author):

It is very clear that the authors took into consideration all of the reviewer comments and worked diligently to improve their manuscript. The manuscript is cohesive, reads well, and presents interesting findings. I have no further comments.